# Difficulty in artificial word learning impacts targeted memory reactivation and its underlying neural signatures

**Arndt-Lukas Klaassen[1,2]\*, Björn Rasch[1]\***

[1]Department of Psychology, Division of Cognitive Biopsychology and Methods, University of Fribourg, Fribourg, Switzerland; [2]Department of Anesthesiology & Pain Medicine, Inselspital, Bern University Hospital, University of Bern, Bern, Switzerland

## eLife assessment

This study provides **useful** findings on how phonetic properties of words, i.e., their difficulty and prior knowledge, influence the outcome of targeted memory reactivation (TMR) during sleep. While these findings are supported by **solid** evidence, they are based on a small sample size warranting future work to shed further light on the impact of TMR in language learning.

**\*For correspondence:**
arndt-lukas.klaassen@unifr.ch
(A-LK);
bjoern.rasch@unifr.ch (BR)

**Competing interest:** The authors declare that no competing interests exist.

**Abstract** Sleep associated memory consolidation and reactivation play an important role in language acquisition and learning of new words. However, it is unclear to what extent properties of word learning difficulty impact sleep associated memory reactivation. To address this gap, we investigated in 22 young healthy adults the effectiveness of auditory targeted memory reactivation (TMR) during non-rapid eye movement sleep of artificial words with easy and difficult to learn phonotactical properties. Here, we found that TMR of the easy words improved their overnight memory performance, whereas TMR of the difficult words had no effect. By comparing EEG activities after TMR presentations, we found an increase in slow wave density independent of word difficulty, whereas the spindle-band power nested during the slow wave up-states – as an assumed underlying activity of memory reactivation – was significantly higher in the easy/effective compared to the difficult/ineffective condition. Our findings indicate that word learning difficulty by phonotactics impacts the effectiveness of TMR and further emphasize the critical role of prior encoding depth in sleep associated memory reactivation.

## Introduction

Learning of new words is essential in language acquisition. During the learning process of unknown words, top-down influences can facilitate perception and encoding of these words depending on the similarity between the new words and the pre-existing knowledge. One critical characteristic of similarity to pre-existing knowledge in auditory word processing is its speech sound (phoneme) pattern. In phonology as the field of language specific phoneme structures, phonotactics determines the constraints of word phoneme composition of a specific language. Previous studies have shown that artificial words with similar phonotactic properties to real words are easier to learn and to remember (*Frisch et al., 2000*; *Gupta and Tisdale, 2009*), suggesting that pre-existing knowledge about the sound structure (i.e. schemas) facilitates encoding and consolidation of new words. A synchronized interplay between the ventromedial prefrontal cortex, hippocampus and unimodal associative cortices is proposed to explain schema instantiation and schema mnemonic effects (*Gilboa and Marlatte, 2017*) at the neural level. To describe the influence of phonotactics on memory processing within this

framework, new words with phoneme structures congruent to activated schema variables are inserted with less cognitive effort and difficulty into pre-existing knowledge networks of long-term memory in contrast to schema-incongruent words.

According to the complementary system account by Gaskell and colleagues (*Davis and Gaskell, 2009*; *McClelland et al., 1995*), learning of new words comprises a first stage of rapid initial familiarization represented by neural activities of the medial temporal lobe and a second stage of slow offline consolidation of the neocortex. The model suggests that, after initial hippocampal learning, through sleep consolidation memory representations, the words are gradually integrated into long-term lexical and phonological memory structures of the neocortex. Additionally, these processes of memory consolidation might depend on spontaneous and repeated reactivation of the newly formed memory traces during non-rapid eye-movement (NREM) sleep (*Rasch and Born, 2013*). The active system consolidation hypothesis proposes synchronized hippocampal sharp-wave-ripples, thalamo-cortical sleep spindles and slow oscillations of NREM sleep as underlying activities of memory reactivation. Accordingly, studies in humans *Mölle et al., 2011*; *Cox et al., 2014*; *Staresina et al., 2015* and in rodent models *Siapas and Wilson, 1998*; *Sirota and Buzsáki, 2005* found that sleep spindles and hippocampal sharp wave ripples occur grouped during slow oscillations and that these events correlate with memory consolidation (*Rasch and Born, 2013*; *Diekelmann and Born, 2010*; *Buzsáki, 1989*; *Girardeau and Zugaro, 2011*; *Singer et al., 2013*).

While similarity of the learning material to pre-existing knowledge clearly facilitates encoding, this notion is controversial for the effects on sleep consolidation. For example, Havas and co-authors (*Havas et al., 2018*) tested learning and consolidation of artificial words phonologically derived either from participant's native language (L1 words in Spanish) or foreign language (L2 words in Hungarian). Here, sleep-associated improvements in memory recognition were only observable in the condition of the L2 words, suggesting that a reduced similarity to pre-existing knowledge facilitates consolidation during sleep. Also, in a study by *Payne et al., 2012*, sleep-associated memory benefits occurred only for semantically unrelated word-pairs, but not for semantically related words.

On the other hand, *Zion et al., 2019* reported that sleep facilitated consolidation in a second language learning task in participants with a higher degree of meta-linguistic knowledge, indicating a beneficial effect of pre-existing knowledge on new word consolidation. Similarly, *Durrant et al., 2015* reported that only schema-conformant but not non-conformant learnings benefitted from sleep. In a study from our own lab, German-participants learning unknown Dutch vocabulary profited more from sleep compared to native French speaker when given the same amount of learning trials, whereas German native speakers might have more pre-existing knowledge about the Dutch words in relation to the French native speakers, which facilitated learning and consolidation of the new Dutch vocabulary (*Cordi et al., 2023*).

A promising approach to examine sleep-associated memory consolidation is to experimentally bias reactivation by re-presenting olfactory (*Rasch et al., 2007*; *Diekelmann et al., 2011*; *Diekelmann et al., 2012*) or auditory (*Rudoy et al., 2009*; *Schreiner and Rasch, 2015*) reminder cues during sleep, a technique known as targeted memory reactivation (TMR). TMR has been related to elevated slow-wave and spindle activity following cueing presentations as neural markers of memory reactivation (*Cairney et al., 2018*; *Antony et al., 2018*). So far, two studies suggest that prior knowledge increase the effectiveness of TMR: Groch and colleagues (*Groch et al., 2017*) reported improved memory performance by TMR for associations between pseudo-words and familiar objects, but not for word associations of new/unknown objects. Here, before sleep, subjects showed as well better encoding performance for the familiar compared to the unknown objects. In another study, *Creery et al., 2015* of learning sound-object locations, the effectiveness of TMR correlated positively with the initial encoding performance prior to sleep. Additionally, both studies *Groch et al., 2017*; *Creery et al., 2015* found significant associations between TMR's effectiveness and sleep spindle activities after the cueing presentations. These findings suggest that higher similarity between the learning material and the pre-existing knowledge facilitates reactivation and consolidation during sleep. However, it is unknown whether these findings can be generalized to learning of new artificial words with different levels of phonotactical similarity to prior word-sound knowledge, as a factor of learning difficulty.

To address this issue, we designed artificial words with varying difficulty levels by phonotactical properties. To consider the beneficial effect of reward related information on sleep-dependent memory consolidation and reactivation (*Asfestani et al., 2020*; *Fischer and Born, 2009*; *Lansink*

*et al., 2009*; *Sterpenich et al., 2021*), we trained healthy young participants to categorize these words into rewarded and unrewarded words to gain and to avoid losses of money points. By using auditory TMR, we re-presented the artificial words during subsequent sleep to study its effect on categorization performance. As a manipulation check, we expect that artificial words with more familiar phonotactical properties would be easier to learn compared to words with less familiarity. Regarding previous findings (*Groch et al., 2017*; *Creery et al., 2015*), we hypothesized higher effectiveness of TMR for the easy to learn words in comparison to the difficult words, accompanied by higher oscillatory activity on the slow-wave and spindle range.

## Results

### Experimental design

To study the impact of difficulty in word learning on TMR, we developed a novel learning paradigm. We formed four sets of artificial words (40 words per set; see *Supplementary files 1 and 2*) consisting of different sequences of two vowels and two consonants. Here, we subdivided the alphabet into two groups of consonants (C1: b, c, d, f, g, h, j, k, l, m; C2: n, p, q, r, s, t, v, w, x, z) and vowels (V1: a, e, I; V2: o, u, y). Four-letter-words were created by selecting letters from the vowel and consonant groups according to four different sequences (G1:C1, V1, V2, C2; G2: C1, V1, C2, V2; G3: V1, C1, C2, V2; G4: V1, C1, V2, C2; *Figure 1a*; see methods for further details). Comparison analyses between the sets revealed significant differences in phonotactic probability (PP; *Figure 1b*; unpaired *t*-tests: G1 / G2>G3/G4, p<0.005, values of Cohen's *d*>0.71). PP quantifies the frequency of a single phoneme (phoneme probabilities) or a sequence of phonemes (e.g. biphone probabilities) within a language and thus serves as a measurement of the similarity between the artificial words and the pre-existing real word knowledge. According to distinct levels of PP, we paired the four sets to the high- (G1 and G2) and low-PP (G3 and G4) condition, respectively.

During encoding, we trained the subjects to discriminate the artificial words based on reward associations by manual button presses. Therefore, the high- and the low-PP condition, consisted of 40 rewarded and 40 unrewarded words. As a two alternative forced-choice task, we assigned left- and right-hand button presses to the rewarded and the unrewarded word category, counterbalanced across subjects. We instructed the participants to respond to each word by left- or right-hand button presses, whereas one button means the word is rewarded (gain of money points) and the other button means the word is unrewarded (avoid the loss of money points). In the beginning, they had to guess. By three presentations of each word in randomized order and by feedback at the end of each trial, they learned to respond correctly according to the rewarded/unrewarded associations of the words (*Figure 1c*).

To obtain a measurement of discrimination memory with respect to the potential influence of the response bias, we applied the signal detection theory (*Green and Swets, 1966*). Because, we instructed the participants to respond to each word by left- or right-hand button presses and that one button means reward is present whereas the other button means reward is absent, we considered correct responses of words of the rewarded category as hits and words of the unrewarded category as correct rejections. Accordingly, we assigned the responses with regard to the reward associations of the words to the following four response types: hits (rewarded, correct); correct rejections (unrewarded, correct); misses (rewarded, incorrect); and false alarms (unrewarded, incorrect). Dependent on responses, subjects received money points (*Figure 1d*).

After encoding and before sleep, we tested the pre-sleep memory performance (*Figure 1e*). Subjects slept one night in the sleep laboratory and during the NREM sleep stages 2 and 3 we conducted auditory TMR of the low-PP words in one group of subjects and TMR of the high-PP words in the other group (between subject design). At the following morning, we tested the post-sleep memory performance.

### Manipulation check based on encoding analyses

To validate our novel paradigm, we examined the influence of PP on encoding performance. Based on the signal detection theory (*Green and Swets, 1966*), we calculated d'-values to measure subject's abilities to differentiate and categorize rewarded and unrewarded words of the two PP conditions over the three presentations of the encoding task (*Figure 2a*). A repeated-measure ANOVA on d' with

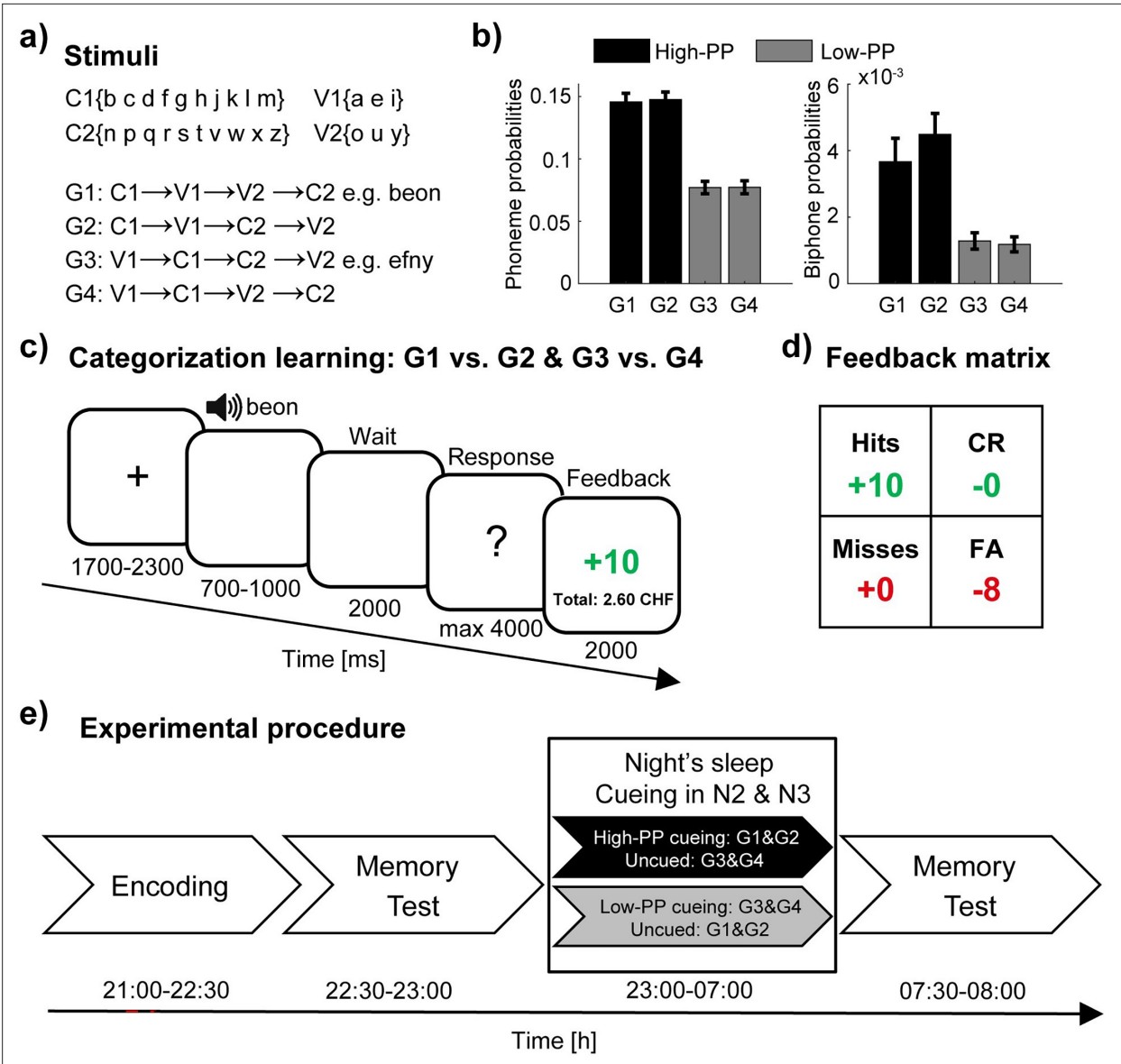

**Figure 1.** Experimental design. (**a**) Artificial words consist of two vowels and two consonants according to four different sequences. (**b**) The phonotactic probabilities (PP) for single phonemes (left panel) and biphone probabilities (right panel) averaged values of the four sets (40 words per set) and pairing of the sets with respect to two distinct levels of PP in high- (black) and low-PP (gray). (**c**) Schematic trial structure of the learning task with screen images and the durations in milliseconds. As a two alternative forced-choice task, responses of left- and right-hand button presses were assigned to the rewarded and the unrewarded word category, respectively. The participants were instructed to respond to each word by left- or right-hand button presses, whereas one button means the word is rewarded (gain of money points) and the other button means the word is unrewarded (avoid the loss of money points). (**d**) Feedback matrix with the four answer types (hits: rewarded and correct; CR, correct rejections: unrewarded and correct; misses: rewarded and incorrect; FA, false alarms: unrewarded and incorrect) regarding to response and reward assignment of the word. Note, subjects could receive and lose money points dependent on correct and incorrect responses. (**e**) Experimental procedure with experimental tasks and phases in temporal order. TMR took place in the NREM sleep stages 2 and 3. Error bars reflect standard errors of the mean (SEM; n=40).

PP (high vs. low) and presentations (1–3) as within-subjects factors revealed significant main effects of PP ($F(1,32) = 5.13$, p=0.03, $\eta^2$=0.14), and presentations ($F(2,64) = 95.67$, p<0.001, $\eta^2$=0.75). Additional pairwise comparisons between PP conditions showed significant differences after the first presentation (paired $t$-tests of the three presentations in order: (1) $t(32) = 0.73$, p=0.47; (2) $t(32) = 2.35$, p=0.03, Cohen's $d$=0.41; (3) $t(32) = 2.17$, p=0.04, Cohen's $d$=0.38). These results indicate superior learning performance for words with high- compared to words with low-PP.

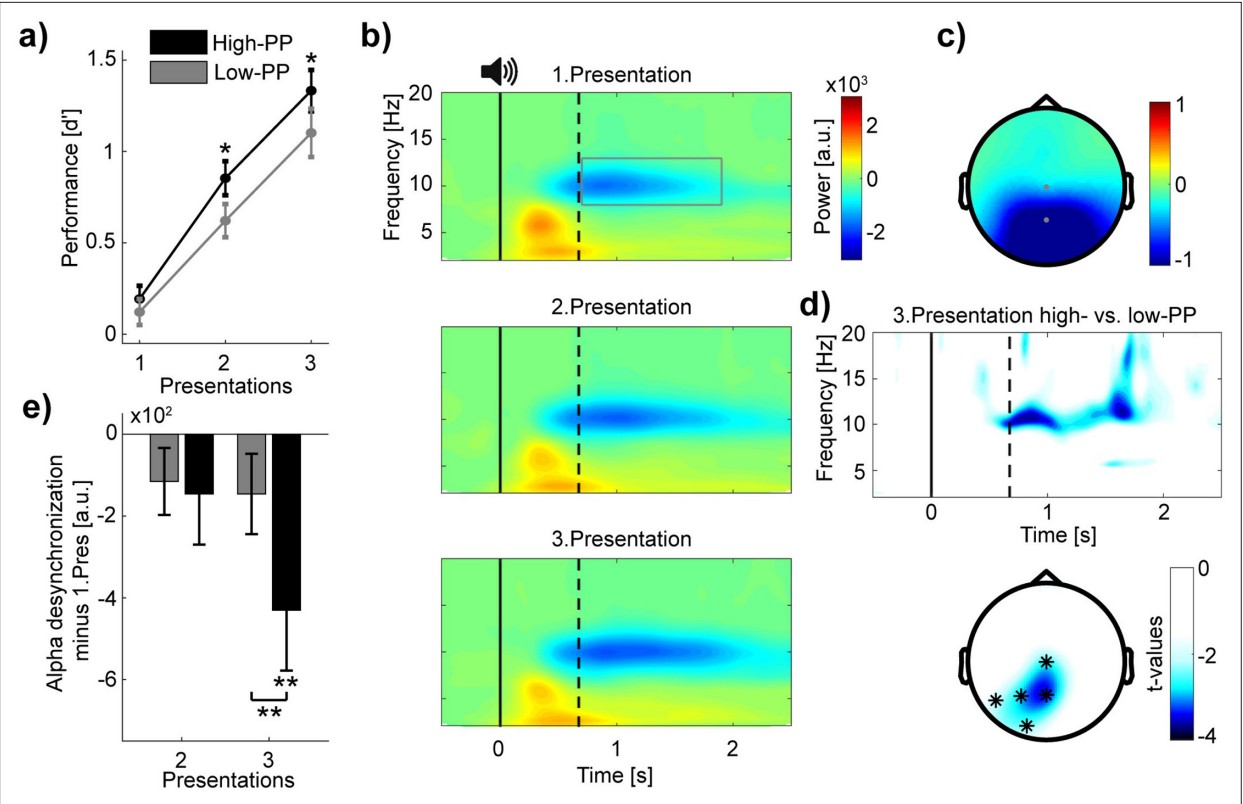

**Figure 2.** Distinct levels of encoding between high- and low-PP words. (**a**) Learning curves showing encoding performance over presentations of high- (black) and low-PP (gray) words. Note significant greater performance of high- in comparison to low-PP over the second and third presentations. (**b**) Grand average time-frequency plots time-locked to word presentations. The gray rectangle within the top panel borders time and frequency range of interest (0.7–1.9 s; 8–13 Hz). Three different panels from top to down regarding to the three presentations. Solid and dotted lines within plots representing stimulus onset and averaged offset, respectively. Note, increases of oscillatory desynchronization in the alpha range (8–13 Hz) over the three presentations. (**c**) Topographic map shows power values averaged over the time window and frequency range of interest. (**d**) (Top) Time-frequency representation of *t*-values (merged over Pz and P3 electrodes) shows significant greater changed desynchronization in alpha oscillations of high- in contrast to low-PP during the third presentation. Below, topographic map indicates significant cluster of electrodes of comparison between PP conditions of the third presentation (0.7–1.9 s; 8–13 Hz). (**e**) The bar chart shows mean changes across subjects in alpha power (merged over Pz and Cz electrodes) of the second and third presentation by subtracting the first presentation in high- (black) vs. low-PP (gray). Statistical analyses revealed significant higher desynchronization of high- compared to low-PP and a significant decrease in alpha power under 0 of high-PP at the third presentation. Error bars reflect standard errors of the Mean (SEM; n=33); *p<0.05, **p<0.01.

To examine the oscillatory correlates of encoding performance, we conducted time-frequency analyses time-locked to auditory word representations (*Figure 2b*) and extracted power values over time, frequency, and EEG electrodes by using wavelet transformations (see methods). This analysis revealed a strong desynchronization after word presentations (0.7–1.9 s to stimulus onset) in the frequency range of the alpha waves (8–13 Hz) and pronounced over parietal and occipital electrodes in contrast to baseline activity (−1 to −0.1 s to stimulus onset; *Figure 2c*). Testing whether changes in alpha desynchronization correspond to increased encoding performance over repetitive presentations, a repeated-measure ANOVA on alpha power values merged over central electrodes (Pz, Cz) including presentations (1–3) as a single within-subjects factor revealed a significant increase in alpha desynchronization ($F(2,64) = 4.19$, p=0.02, $\eta^2$=0.12).

Regarding distinct levels of encoding performance between high- and low-PP, we compared next changes in alpha desynchronization between these conditions. We obtained the changes in power values by subtracting the first from the second and third presentation for the high- and low-PP condition, respectively. Here, the first word presentation of naive stimulus processing served us with a more representative baseline condition covering the time-window of interest of 0.7–1.9 s after the stimulus onset to examine relevant changes of encoding. Cluster-based statistics of the third presentation revealed a significant cluster over the left posterior electrodes with a stronger alpha power decrease

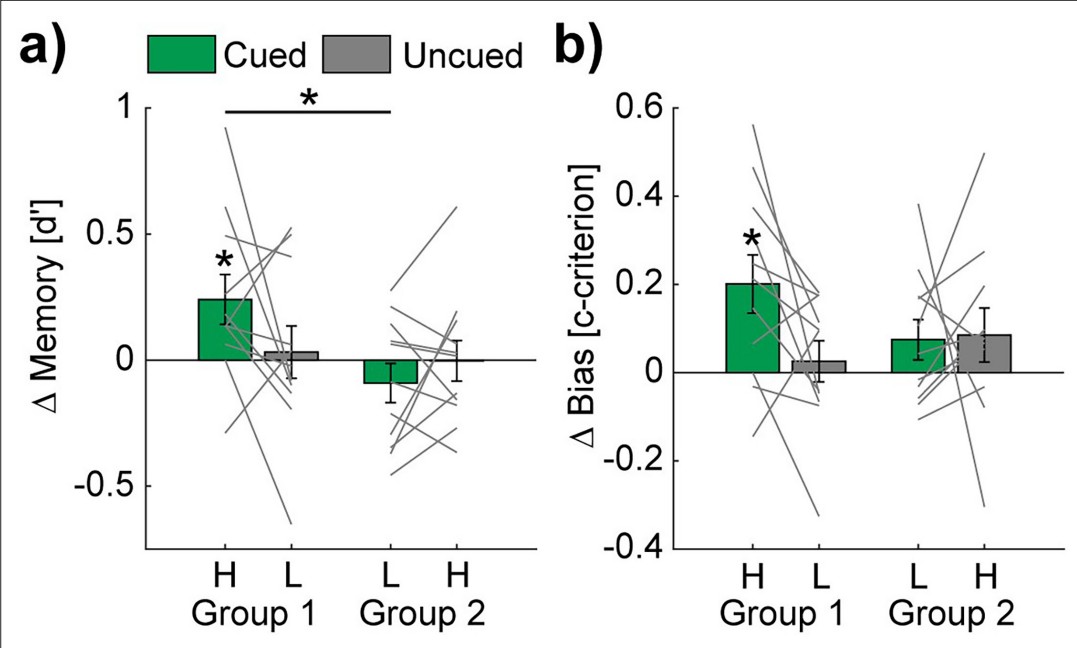

**Figure 3.** TMR affects the memory performance of the easy to learn words. Bar charts show mean overnight changes of d' (**a**) and c-criterion (**b**) values of high (H) - and low (L) - PP and cued (green) vs. uncued (gray) conditions. Note, statistical analyses revealed significant overnight increases only in the high-PP cued condition. Gray lines represent individual data points. Error bars reflect SEM (n=11); *p<0.05.

The online version of this article includes the following figure supplement(s) for figure 3:

**Figure supplement 1.** Pre- and post-sleep memory test data.

for the high- in contrast to the low-PP condition (time window: 0.7–1.9 s; frequency range 8–13 Hz; averaged *t*-value over the five cluster electrodes = –2.69, p=0.01, Cohen's *d*=–0.48; *Figure 2d*). A post-hoc repeated-measure ANOVA on alpha power changes (merged over Pz and Cz electrodes) with PP (high vs. low) and presentations (2–3) as within-subjects factors revealed a main effect of PP ($F(1,32)$ = 5.42, p=0.03, $\eta^2$=0.15), and a significant interaction ($F(1,32)$ = 7.38, p=0.01, $\eta^2$=0.19; *Figure 2e*). Additional post-hoc pairwise comparison between PP conditions showed significant difference of the third presentation (paired *t*-tests of presentations in order: (2) $t(32)$ = –0.36, p=0.72; (3) $t(32)$ = –3.55, p=0.001, Cohen's *d*=–0.62) and a *t*-test against 0 revealed significantly decreased activity in high-PP of the third presentation ($t(32)$ = –2.92, p=0.006, Cohen's *d*=–0.51). In addition to the behavior results, these EEG results indicate differences between PP conditions in desynchronization of alpha oscillations, as an assumed neural correlate of encoding depth (*Hanslmayr et al., 2009*; *Griffiths et al., 2021*). To summarize, as a manipulation check based on encoding analyses, we confirmed that the conditions of high- and low-PP correspond to distinct levels in learning difficulty.

## TMR affects memory consolidation of the easy to learn words

To examine whether TMR during sleep impacts memory consolidation of discrimination learning with respect to learning difficulty, we calculated the overnight changes by subtracting the pre- from the post-sleep memory performance based on d'-values of the reactivated sequences (cued) and non-reactivated sequences (uncued). In group 1, the high-PP sequence was cued, and the low-PP sequence was uncued. In group 2, the low-PP sequence was cued, and the high PP sequence was uncued (see *Figure 3—figure supplement 1* for pre- and post-sleep memory test data of the different conditions). Conducting *t*-tests against 0 revealed a significant increase in the TMR/cueing condition of group 1 (high-PP cued: 0.24±0.1, $t(10)$ = 2.44, p=0.035, Cohen's *d*=0.74), while the memory performance remained unchanged in all the other conditions (low-PP cued: –0.09±0.08, $t(10)$ = –1.16, p=0.27; high-PP uncued: 0±0.08, $t(10)$ = –0.04, p=0.97; low-PP uncued: 0.03±0.1, $t(10)$ = 0.31, p=0.76; *Figure 3a*).

An additional two-way mixed design ANOVA on the same values with the factor cueing (cued vs. uncued) as a within-subject factor and group as a between-subject factor revealed trends of significance (p<0.1) for the interaction (cueing ×group: $F_{(1,20)}$ = 3.47, p=0.08) and the main effect of group ($F_{(1,20)}$ = 3.28, p=0.09). The main effect of cueing was not significant ($F_{(1,20)}$ = 0.58, p=0.46). Post-hoc pairwise comparisons revealed a significant difference between the conditions of high-PP-cued and low-PP-cued ($t_{(20)}$ = 2.63, p=0.02, Cohen's $d$=1.12), a trend of significance between high-PP-cued and high-PP-uncued ($t_{(20)}$ = 1.91, p=0.07) and no significant difference between high-PP-cued and low-PP-uncued ($t_{(10)}$ = 1.55, p=0.15). In additional control analyses, pre-sleep memory performance and vigilance shortly before the post-sleep memory task did not significantly (*p-values* ≥0.08) differ between both cueing groups (see *Supplementary file 3*).

We examined next as an exploratory analysis whether TMR conditions influence biases in decision-making. Therefore, we analyzed changes between the pre- and post-sleep memory tasks of c-criterion values as a measurement of risk avoidance/seeking in decision making (*Green and Swets, 1966*). Here, we found as well a significant overnight increase for cued sequences in group 1 (high-PP cued: 0.2±0.07, $t_{(10)}$ = 3.04, p=0.01, Cohen's $d$=0.92), indicating an increased bias towards risk avoidance by TMR (*Figure 3b*), while all the other conditions remained unchanged (high-PP uncued: 0.09±0.06, $t_{(10)}$ = 1.4, p=0.19; low-PP cued: 0.08±0.05, $t_{(10)}$ = 1.63, p=0.13; low-PP uncued: 0.03±0.05, $t_{(10)}$ = 0.56, p=0.59). An ANOVA on c-criterion changes showed no significant effects (interaction cueing ×group: $F_{(1,20)}$ = 2.66, p=0.12; main effect cueing $F_{(1,20)}$ = 2.08, p=0.17; main effect group $F_{(1,20)}$ = 0.38, p=0.55).

Taken together, these results suggest that the effectiveness of TMR depends on the level of difficulty in word learning, while auditory cueing during sleep increases the memory performance of the easy to learn words.

## Increased spindle power nested during slow wave up-states in TMR of the easy to learn words

After analyzing TMR's effectiveness on behavior, we investigated the corresponding neural activities by EEG. By visual inspection of the signals, auditory word presentations during NREM sleep led to broad high-amplitude oscillations, called slow waves (SW; 0.5–3 Hz), whereas sleep spindle activity (9–16 Hz) with various amplitude occurred preferentially nested during the SW's up-state phase (see for example traces *Figure 4a*).

To statistically analyze whether TMR increases the SW density, we conducted a detection algorithm. To control for individual differences, we included 30% of SW with the highest amplitude per subject (see methods) in subsequent analyses. Testing changes of SW density in time windows of 0.5 s, revealed significant increases after stimulus onset in comparison to baseline (*Figure 4b*; baseline period of –1.5–0 s before stimulus onset; $t$-tests against 0; 0–0.5 s: 49.73 ± 14.32%, $t_{(21)}$ = 3.47, p=0.002, Cohen's $d$=0.74; 0.5–1 s: 83.44 ± 14.97%, $t_{(21)}$ = 5.57, p<0.001, Cohen's $d$=1.19; 1.5–2 s: 40.33 ± 10.67%, $t_{(21)}$ = 3.78, p=0.001, Cohen's $d$=0.84; see *Supplementary file 4* for all statistical comparisons of time-bins from –0.5–3 s and *Figure 4—figure supplement 1* for SW distribution from –0.5–6 s time-locked to stimulus onset). Additional comparisons between the TMR conditions of high- and low-PP showed no significant differences in SW density, SW amplitude, number of TMR presentations and other sleep parameters, but a trend of significance for REM sleep parameters (see *Supplementary file 5*). In sum, these results indicate an increase of SW in response to auditory word stimulation independent of difficulty in word learning.

Next, we analyzed whether variations in spindle-band oscillations nested during the SW up-states after auditory TMR – as a neural signature of memory reactivation – differ between the conditions of high- and low-PP. To extract the spindle power during SW up-states, we conducted time-frequency analyses time-locked to the troughs of the detected SW in the time-window from 0 to 6 s after auditory presentations. *Figure 4c* shows the grand average analyses per condition, whereas the fast spindle band activity (12–16 Hz) nested during SW up-states displayed a topography with maximum power over the central-parietal electrodes. Comparison between PP conditions revealed a significant greater spindle-band power prominent in the frequency range of 11–14 Hz during SW up-states (from 0.3 to 0.8 s after the SW trough) of high- in contrast to low-PP (averaged $t$-value over the 5 cluster electrodes = 2.66, p=0.01, Cohen's $d$=1.13; *Figure 4d*), whereas post-hoc comparison including the individual SW amplitude as a covariate still revealed the significance of this cluster ($t_{(20)}$ = 2.38, p=0.03, Cohen's $d$=1.02).

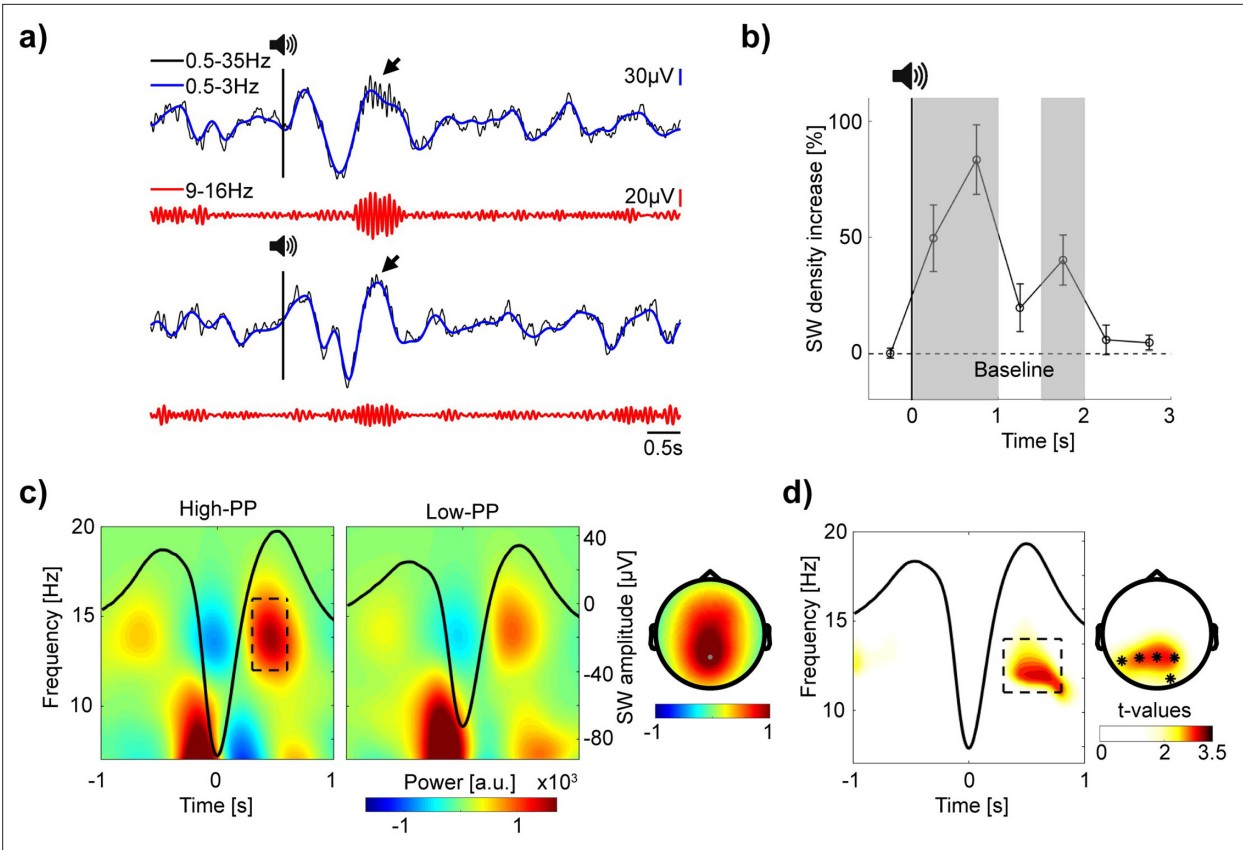

**Figure 4.** Increased spindle power during SW up-states in TMR of the easy to learn words. (**a**) Top and bottom panel, two example EEG traces of auditory cueing during sleep (–2 until 6 s to stimulus onset). Top rows, in blue, signal filtered in the SW range (0.5–3 Hz) superimposed upon the broadband (0.5–35 Hz) signal in black. Vertical black lines with speaker symbols on top mark onsets of auditory presentations. Black arrows point to spindle activity during SW up-states. Bottom rows, in red, the same signal, but filtered in the spindle range (9–16 Hz). Note, elevated SW following cueing presentations with various spindle band activity nested during SW up-states. (**b**) Grand average baseline corrected curve of increased SW density after TMR in percentage. Shaded gray areas mark time windows (0–0.5 s, 0.5–1 s and 1.5–2 s) of significant increased SW density. Error bars reflect SEM (n=22). (**c**) Grand average time-frequency plots time-locked to the troughs of SW with averaged signals plotted as black lines. Two different panels (left and right) according to high- vs. low-PP cueing conditions. The rectangle within the left panel borders time (0.3 until 0.6 s to SW troughs) and frequency range of up-state fast spindle band activity (12–16 Hz). Corresponding topographic map at right shows elevated fast spindle power over mid-parietal electrodes. (**d**) Time-frequency representation of *t*-values time-locked to SW shows significant greater spindle band power during SW up-states for high- vs. low-PP (merged over Pz, P3, and P4 electrodes). Right, topographic map of *t*-values shows corresponding significant cluster of electrodes (0.3–0.8 s; 11–14 Hz), *p<0.05.

The online version of this article includes the following figure supplement(s) for figure 4:

**Figure supplement 1.** Slow wave density distribution.

**Figure supplement 2.** Detected sleep spindle analyses.

By conducting an additional analyses based on detection of fast sleep spindles (12–16 Hz; see Methods), we confirmed that fast sleep spindles during the SW up-states (from 0.3 to 0.8 s after the SW trough) occurred with significantly higher amplitude after the cueing presentation of high- compared to low-PP words, whereas parameters of sleep spindle density and the amount sleep spindles coupled to the SW up-state did not differed between the cueing conditions (see *Figure 4—figure supplement 2* and *Supplementary file 6*).

These results suggest that successful TMR of the easy to learn words is associated with increased spindle activity nested during SW up-states.

Whereas we found a significant group difference in spindle power nested during SW up-states, conducting further whole sample (n=22) correlation analyses between the individual spindle power values of the significant cluster and the overnight changes of behavior measurements revealed no significant correlations (Δ d': *r*=0.16, p=0.48; Δ c-criterion: *r*=0.19, p=0.40).

## Discussion

In this study, we asked whether learning difficulty of artificial words affects TMR's effectiveness and its neural activities. After confirming easier encoding of artificial words with high- compared to low phonotactic familiarity, we found that auditory TMR during sleep improved the memory performance of the easy to learn words, whereas cueing of the difficult words had no effect. Correspondingly, at the neural level, we observed that cueing presentations of the easy words induced significant greater spindle-band oscillations nested during SW up-states in comparison to the difficult words. To our knowledge, we present here the first study to address the critical role of learning difficulty by experimental manipulations of phonotactical word properties on TMR's effectiveness and its neural signatures.

Related to our finding of TMR's effectiveness of the easy words, previous research in object association learning showed beneficial influences of encoding depth *Creery et al., 2015* and prior knowledge *Groch et al., 2017* on TMR. Language studies without using TMR revealed that prior knowledge affected sleep dependent memory integration of new words in children *James et al., 2017* and in adults (*Zion et al., 2019*; *Cordi et al., 2023*). To explain these critical influences on consolidation during sleep, the model of the complementary systems approach of word learning (CLS; *Davis and Gaskell, 2009*) proposes that new word memory temporary encoded in the hippocampus transfer during sleep into cortical structures of lexical and phonological memory for long-term embedding, due to co-activation of both memory systems. Thus, word memory reactivation and consolidation during sleep depend on the hippocampal encoding depth and the cortical prior knowledge in phonotactics. Regarding our results, we suggest facilitated systems co-activation/consolidation of the easy to learn words by auditory cueing during sleep as a manipulation of both factors – via enhanced initial encoding and higher similarity to pre-existing phonotactical knowledge – compared to the difficult to learn words.

Contrary to our results, other studies found better memorization of words with low phonological *Havas et al., 2018* and semantic (*Payne et al., 2012*) similarity to pre-existing knowledge after a retention period of sleep compared to words with high similarity. To explain this discrepancy, we could speculate that a very high degree of similarity led to a strong memory formation during wake-encoding, which prevented an additional effect of consolidation during sleep. Indeed, *Creery et al., 2015* showed that learned object-location associations only benefited from TMR during sleep when not almost perfectly recalled during prior wake. Here, a moderate encoding depth might benefit most from subsequent sleep consolidation compared to insufficient or almost perfect encoding. Alternatively, the contrary results could be related to characteristics of the learning paradigms (L1- vs. L2-like novel words with associations to objects *Havas et al., 2018*; semantically related vs. unrelated word pairs *Payne et al., 2012*) distinct to our design.

By conducting an additional exploratory analysis, we observed a significant change of the decision bias in the cueing condition of the easy to learn words and no overnight changes in the other conditions. Here, subjects became more biased in their decision making to avoid losses of money points. This observation suggests that TMR not only captures specific memory traces of the words and their reward associations, but also decision-making tendencies. In a previous study, *Ai et al., 2018* revealed altered reward-relevant decision preferences by TMR. To what extent concepts of decision-making such as risk avoidance/seeking could be reactivated, transformed, and consolidated by TMR is largely unexplored and might be interesting for future research for example in neuroeconomics.

Correspondent to our behavior results of TMR's effectiveness, we found significantly higher spindle-band oscillations nested during SW up-states after cueing presentations of the easy in comparison to the difficult words. Accordingly, the active system consolidation hypothesis proposes slow wave – sleep spindle coupling activities as an underlying mechanism of systems consolidation (*Rasch and Born, 2013*; *Diekelmann and Born, 2010*; *Klinzing et al., 2019*). In addition to our result of the significant group difference, we failed to find significant correlations between SW nested spindle power values and overnight changes in behavior measurements, whereas previous studies reported associations of SW and spindle activities during sleep with the integration of new memories in pre-existing knowledge networks (*Tamminen et al., 2010*; *Tamminen et al., 2013*). Strikingly, recent studies by implementing machine learning for EEG decoding revealed that classification accuracy of distinct memory categories peaked above chance level synchronized with SW-spindle coupling events (*Cairney et al., 2018*; *Schreiner et al., 2021*). Here, our results support the assumption of SW-spindle

activities as an underlying mechanism of memory reactivation during sleep by providing additional evidence to link successful TMR on behavior to increased spindle activity during SW up-states.

In validation of our novel paradigm, we confirmed easier encoding and better memorization of artificial words with high- in comparison to low-PP, in line with previous studies (*Frisch et al., 2000*; *Gupta and Tisdale, 2009*). These results suggest that participants used their prior knowledge of frequently occurring word-sound patterns to facilitate learning. Regarding the framework of schema theory (*Gilboa and Marlatte, 2017*), processing of incoming words with high-PP might co-activate consistent schemata of prior word sound knowledge and thus may lead to facilitated memory integration and stabilization during encoding in comparison to processing of more inconsistent words with low-PP.

Our EEG analyses of encoding revealed desynchronization of alpha oscillations as a function of learning improvements. Additionally, superior encoding of the high-PP words corresponded to greater alpha desynchronization in comparison to the low-PP words. Consequently, we assume that the suppression of alpha oscillations over parietal and occipital electrodes represent a neural correlate of mnemonic functioning. Accordingly, previous studies linked desynchronization of alpha oscillations to memory processing *Hanslmayr et al., 2009*; *Klimesch et al., 2006* and memory replay during wake (*Michelmann et al., 2016*). A study of intracranial electrophysiology in epilepsy patients (*Griffiths et al., 2019b*) showed that neocortical alpha/beta desynchronization precede and predict fast hippocampal gamma activity during learning. The authors propose that these coupled activities represent hippocampal-cortical information transfer important for memory formation. Here, our findings let suggest cortical alpha desynchronization as neural marker of wake mnemonic processing according to distinct levels of difficulty in word learning. Additional studies linked alpha desynchronization to cognitive effort and cognitive load (*Proskovec et al., 2019*; *Zhu et al., 2021*). So, one could assume to observe higher alpha desynchronization in the more difficult to learn condition of low-PP compared to high-PP. On the other hand numerous studies investigating oscillatory correlates of learning and memory showed that alpha desynchronization is associated with memory across different tasks, modalities and experimental phases of encoding and retrieval (*Hanslmayr et al., 2009*; *Griffiths et al., 2021*; *Michelmann et al., 2016*; *Griffiths et al., 2019b*; *Griffiths et al., 2016*; *Griffiths et al., 2019a*). Strikingly, *Griffiths et al., 2019a* revealed by simultaneous EEG-fMRI recordings a negative correlation between the occurrence of patterns of stimulus-specific information detected by fMRI and cortical alpha/beta suppression. Here, the authors suggested that a decrease of alpha/beta oscillations might represent the neuronal mechanism of unmasking the task-critical signal by simultaneous suppression of task-irrelevant neuronal activities to promote information processing. Following this interpretation, we assume that over the course of learning elevated memory processing of the easier to learn stimuli is associated with enhanced information processing and thus accompanied by higher cortical alpha desynchronization in comparison of the more difficult to learn stimuli.

Our study has limitations due to a small sample size and between-subject comparisons. The criteria of data analyses were not pre-registered and the *p*-values of our behavior analyses were not corrected for multiple comparisons. Further, potential influences of phonotactic word properties and the prior encoding depth on TMR cannot be interpreted independently because the conditions of phonotactic word properties impacted already the encoding depth before sleep. To disentangle these potential factors, future studies could include a higher number of encoding trials for the low-PP words to equalize the encoding depth between conditions prior to TMR, as successfully shown in a recent study (*Cordi et al., 2023*). Additionally, a within-subject design with a larger sample size would provide a more robust control of interindividual differences in sleep and cognition. To control for potential confounders despite the influence of difficulty in word learning on TMR, we compared parameters of sleep, the pre-sleep memory performance and the vigilance shortly before the post-sleep memory test, revealing no significant group differences (see *Supplementary files 3 and 5*). Nevertheless, we cannot rule out that other individual trait factors differed between the groups, such as the individual susceptibility to TMR. To rule out these alternative explanations based on individual factors, we suggest for future research to replicate our study by conducting a within-subject design with cueing of subsets of previously learned low- and high-PP words providing all conditions within the same individuals as shown in other TMR studies (*Schreiner and Rasch, 2015*; *Cairney et al., 2018*). Further, we used artificial words based on American English in combination with German speaking participants, whereas language differences of pronunciation and phoneme structures might affect word perception

and memory processing (*Bohn and Best, 2012*). On the other hand, both languages are considered to have the same language family (*Eberhard et al., 2019*) and the phonological distance between English and German is quite short compared for example to Korean (*Luef and Resnik, 2023*). Thus, major common phonological characteristics across both languages are still preserved. In addition, our behavior analyses revealed robust word discrimination learning and distinct memory performance according to different levels of phonotactic probabilities providing evidence of successful experimental manipulation. Based on our paradigm, we investigated categorization learning of artificial words according to their reward associations (rewarded vs. unrewarded) and did not studied aspects of generalization learning of artificial grammar rules (*Reber, 1967*; *Batterink and Paller, 2017*; *Batterink et al., 2014*). This difference might limit the comparability between these previous language-related studies and our findings. However, the usage of artificial words with distinct phonotactical properties provided a successful way to manipulate learning difficulty and to investigate word properties on TMR, whereas our reward categorization learning paradigm had the advantage to increase the relevance of the word learnings due to incentives.

To conclude, our present study demonstrates that difficulty in word learning as a manipulation of phonotactic properties impacts the effectiveness of TMR. Here, TMR of the easy to learn words facilitates categorization learning while TMR of the difficult words had no effect. Auditory cueing of the easy words induced increased spindle activity during the SW up-states – as an assumed underlying activity of memory reactivation – compared to the difficult words. Thus, our findings suggest that future research and clinical applications to restore language capabilities should consider learning difficulty based on phonotactical word properties as a potential factor of successful TMR, whereas alpha desynchronization during initial learning – as a neural marker of encoding depth – may serve as a predictor for sleep-associated memory reactivation and consolidation.

## Methods
### Subjects
The study included 39 subjects (29 females) with an age range of 19–28 years (M=22.28 years, SD = 2.04). Participants were recruited from the University of Fribourg community by E-Mail or through advertisements at the campus of the University. Before participation, subjects had to give written informed consent as approved by the Ethical Commission of the Department of Psychology of the University of Fribourg. All participants were German speakers, and no subject had a history of neurological or psychiatric illness. The participants were instructed to keep a normal sleep schedule, to get up in the morning before 8 a.m. and not to consume alcohol and caffeine on experimental days. For participation, subjects either received credit for an undergraduate class and/or monetary compensation.

Data were excluded from subjects who did not reach the minimal learning performance of d'>1.05 during the pre-sleep memory test in at least one of the two PP conditions, whereas this threshold value corresponds to accuracy rates of 70% (n=5). In addition, we excluded one subject who showed a negative d' in one PP condition of the pre-sleep memory test (n=1).

### Pre-learning
Participants arrived at the sleep laboratory at 19:30 hr. Electrodes for standard polysomnography (32 EEG electrodes, EMG and ECG electrodes) were mounted with two EEG mastoid electrodes. One electrode under the right eye was attached to record eye movements.

### Encoding task
The encoding task started around 21:00 hr. Subjects learned to discriminate rewarded and unrewarded artificial words by right- and left-hand button presses. As a two alternative forced-choice task, we assigned left- and right-hand button presses to the rewarded and the unrewarded word category, counterbalanced across subjects. We instructed the participants to respond to each word by left- or right-hand button presses, whereas one button means the word is rewarded (gain of money points) and the other button means the word is unrewarded (avoid the loss of money points). During each trial, an initial fixation cross was displayed between 1.7 and 2.3 s. Subsequently, concurrent with the onset of a blank screen the sound of an artificial word was played for 0.7–1 s. After a waiting

period (blank screen 2 s), a question mark appeared to signal the onset of the response time window (maximal duration 4 s). Following the key press response, a feedback screen with the money points of the trial and the current task score appeared for 2 s. The learning task contained 480 trials in randomized order according to 160 artificial words with three presentations each.

### Pre-sleep memory task

After finishing the encoding task, following a 10 min break, subjects performed the pre-sleep memory task. The memory task had the same trial structure as the encoding task without the last feedback screen. Participants received again money points for correct responses. However, the received amount was only shown at the end of the task. The memory task contains 160 trials according to the 160 artificial words.

### Auditory target memory reactivation during sleep

Following an additional impedance check and re-adjustment of the EEG electrodes, subjects went to bed in a noise and electric shielded cabin of the sleep laboratory. All night sleep periods started with light off between 23:00 and 24:00 hr. Based on online monitoring of N2 and N3 sleep, artificial words were presented aurally via loudspeakers (sound pressure level 55 dB) with a randomized inter-stimulus interval of 8±2 s. One group of subjects were exposed to artificial words with low phonotactic probability during sleep, whereas we presented to the other group the high phonotactic probability words (see *Supplementary file 5* for number of reactivations). We interrupted word presentations when we observed online an arousal or patterns of REM sleep.

### Post-sleep memory task

After sleep and re-adjustment of the EEG, subjects performed the post-sleep memory task (see above the description of the pre-sleep memory task).

### Experimental tasks

All experimental tasks, including sleep reactivation, were conducted by using the E-Prime software (Psychology Software Tools, Sharpsburg, USA). By presentation of practice trials at the beginning, the subject's understanding of the task was approved. Across all tasks, stimuli were presented in randomized order.

### Stimuli

To create artificial words, we subdivided the alphabet in two groups of consonants (C1: b, c, d, f, g, h, j, k, l, m; C2: n, p, q, r, s, t, v, w, x, z) and vowels (V1: a, e, I; V2: o, u, y). Four-letter-words were created by selecting letters from the vowel and consonant groups according to four different sequences (G1:C1, V1, V2, C2; G2: C1, V1, C2, V2; G3: V1, C1, C2, V2; G4: V1, C1, V2, C2). Artificial words were converted automatically from text to speech files (wav format) by using MATLAB functions (tts.m, audiowrite.m) with the setting of a female computer voice with American English pronunciation. From this pool of 900 stimuli for each rule, we selected 40 artificial words per rule category (4×40 words) according to the inclusion criteria: fluent and understandable pronunciation; no meaning; no names; aurally discriminable to other selected words (see *Supplementary files 1 and 2* for lists of the used words).

Phonotactic probability values were computed for each word by using an online computation platform (*Aljasser and Vitevitch, 2018*) (https://calculator.ku.edu/phonotactic/English/words) according to an American English lexicon. Between rule category comparison analyses revealed significant differences in phonotactic probabilities between two rules in contrast to the other two rules (see *Figure 1b*).

### Psychomotor vigilance task

With respect to vigilance after sleep before the post-sleep memory task, a reaction time measurement was conducted by the Psychomotor Vigilance Task (PVT) (*Dinges and Powell, 1985*). Each trial began with a fixation cross of a randomized duration between 2 and 10 s. Participants were instructed to press the spacebar button with the forefinger of the non-dominant hand as quickly as possible after

numbers started to count in milliseconds on the screen. The reaction time was displayed for 1 s after the key press. The PVT had a duration of 10 min (see for results *Supplementary file 3*).

## Behavior analyses

As a measurement of discrimination learning and memory performance, we calculated d'-values (d'=z(hits)-z(false alarms)) according to the signal detection theory (*Green and Swets, 1966*). Here, we considered correct trials with gaining of money points as hits and incorrect trials with a loss of money points as false alarms (see *Figure 1d*). In addition, we assessed measurements of the response bias (c=−0.5*(z(false alarms)+z(hits))). According to the convention (*Macmillan and Kaplan, 1985*), rates of 0 and 1 were replaced with 0.5 /n and with (n-0.5)/n respectively, whereas n corresponds to the number of trials (n=40). Values of d' and response bias were calculated separately for the low- and high-PP condition.

## EEG recordings

We made EEG recordings by using customized 32-Ag/AgCl electrodes at 10–10 locations caps (EASYCAP, Woerthsee-Etterschlag, Germany) and 32 channel amplifiers (Brain Products, Gilching, Germany). Impedances were kept below 10 kΩ. The EEG was recorded with a sampling rate of 500 Hz using Brain Vision Recorder software (Brain Products, Gilching, Germany). Signals were referenced to electrodes at the mastoids. The ocular activity was measured via one EOG channel mounted ~2 cm below the right eye. Muscle tone was monitored by EMG recordings made under the chin. The following EEG electrodes were used for subsequent wake and sleep analyses: Fp1, Fp2, F3, F4, C3, C4, P3, P4, O1, O2, F7, F8, T7, T8, P7, P8, Fz, Cz, and Pz.

## Sleep scoring

The sleep stages of NREM 1–3 (N1 to N3), wake, and REM sleep were scored offline and manually according to the criteria of the American Academy of Sleep Medicine (AASM) by visual inspection of the signals of the frontal, central, and occipital electrodes over 30 s epochs (*Iber et al., 2007*). Based on offline scoring, we confirmed TMR exposure during N2 and N3 and no significant differences (p-values >0.05) of sleep parameters between the cueing groups (see *Supplementary file 5*).

## Preprocessing wake EEG

Wake EEG were preprocessed using the Fieldtrip toolbox (http://fieldtriptoolbox.org; Donders Institute for Brain, Cognition and Behaviour, Radboud University, Netherlands) (*Maris and Oostenveld, 2007*). First, raw data signals were re-referenced to averaged mastoid electrodes and filtered (high- and low-pass, 0.5 and 45 Hz). For each trial, data were segmented into epochs (−2–4 s) time-locked to stimulus onset. After demean and detrend processing steps, noisy trials were identified by visual inspection and discarded from further analyses. We then conducted an independent component analysis (ICA) to identify and reject ICA components impacted by eye blinks and eye movements.

## Preprocessing sleep EEG

EEG signals during sleep were re-referenced to averaged mastoid electrodes and filtered (high- and low-pass, 0.5 and 35 Hz). To analyze EEG activities of TMR, data were segmented into epochs (−2–8 s) around the onset of stimuli presentations. After demean and detrend processing, trials with signals distorted by noise and movement artefacts were identified by visual inspection and removed from further analyses.

## Time-frequency analysis wake data

We conducted the time-frequency analyses by using Morlet wavelet analyses implemented in the Fieldtrip toolbox. The number of wavelet cycles was adjusted to seven cycles. We extracted oscillatory power in the frequency range of 1–45 Hz with frequency steps of 0.2 Hz and time steps of 10ms. All the power values were baseline corrected and transformed to absolute changed values by subtracting the corresponding averaged values of the baseline interval of –1 to –0.1 s before stimulus onset.

## Slow wave-spindle power analyses

To reduce influence of potential confounders on TMR like low prior learning performance and number of reactivations, we included only subjects with a pre-sleep memory performance d'>0.75 in both PP

conditions and who did receive a minimal number of 160 reactivations (each word was represented at least two times during sleep). Thus, we excluded 5 and 6 subjects, respectively. The final sample for sleep analyses consisted of n=22 with 11 subjects in each cueing condition.

To detect slow waves, we first localized all negative and positive peaks of the 0.5–3 Hz band-pass filtered signal during the auditory cueing trials in the time windows of 0–6 s according to stimulus onsets. In addition, time distance between the prior and posterior positive peaks should be in the range of 0.3–2 s. For the following analyses, the 30% slow waves with the highest amplitude (from negative peak to posterior positive peak) were included for each subject.

We then extracted the corresponding spectral power (1–30 Hz) in a time window around this negative slow wave peak (±2 s) using Morlet wavelet (7 cycles) analysis implemented in the Fieldtrip toolbox, with a time window and step size of 10ms and frequency steps of 0.2 Hz. Extracted power data over frequency and time was averaged of the slow waves and baseline corrected by subtracting the averaged values of the baseline interval of –2 to –1.5 s prior to the negative slow wave peak. Finally, the transformed absolute changed values were plotted for the frequency and time range of interest and different conditions (see *Figure 4c*).

## Slow wave density analyses

To analyze the number of slow waves around memory reactivations during sleep, we divided the time course of reactivation trials in equally sized bins with a duration of 0.5 s from –1.5–3 s according to stimulus onset. Based on negative peaks of detected slow waves, we counted the occurring of slow waves for each bin over trials. Subsequently, we computed density values for each bin by division of the number of trials. Finally, we calculated the relative increase in slow wave density in percent by division of the averaged baseline values (bins: –1.5–0 s; see *Figure 4b*). In an additional histogram, we show the grand average slow wave density distribution in number per trial without baseline correction and with a bin size of 0.5 s from –0.5 to 6 s time-locked to stimulus onset (see *Figure 4—figure supplement 1*).

## Sleep spindle analyses

We detected fast sleep spindles by band-pass filtering (12–16 Hz) the signal of the Pz electrode during the auditory cueing trials in the time windows of –2–8 s according to stimulus onsets. The amplitude threshold was calculated individually for each subject as 1.25 standard deviations (SDs) from the mean. The beginning and end times of the sleep spindles were then defined as the points at which the amplitude fell below 0.75 SDs before and after the detected sleep spindle. Only sleep spindles with a duration of 0.5–3 s were included in subsequent analyses.

To compare the sleep spindle densities between the different cueing conditions of high- and low-PP, we computed the grand average sleep spindle density distribution in number per trial with a bin size of 0.5 s from –0.5 to 6 s time-locked to stimulus onset in each condition (see *Figure 4—figure supplement 2a* and *Supplementary file 6*).

Based on the detected slow waves and sleep spindles, we defined coupling events when the positive amplitude peak of a detected sleep spindle was occurring during the slow wave up-state phase in a time window of 0.3–0.8 s according to the trough of a slow wave.

We computed the averaged amplitude size of each detected sleep spindle by calculating the mean of the absolute amplitude values of all negative and positive peaks within a detected sleep spindle (see *Figure 4—figure supplement 2b*).

## Statistical analysis

To test for differences between conditions of wake behavioral data, repeated-measure ANOVAs and paired sample *t*-tests were used. To analyze overnight changes of sleep behavioral data within TMR conditions, we conducted at first dependent sample *t*-tests against 0 of Δ-values (post-sleep test minus pre-sleep test) of d' and c-criterion (see *Figure 3*). Two-way mixed design ANOVAs were computed to compare Δ-values between TMR conditions. After confirming at least a trend of significance (p<0.1) for the interaction effect, we conducted post-hoc pairwise comparisons by independent and dependent sample *t*-tests. For all behavior statistical analyses, the p-value was set at p<0.05 for two-tailed testing. A p-value <0.1 and>0.05 was reported as a trend of significance.

All statistical analyses were processed by using MATLAB 2018a (MathWorks, Natick, USA) and SPSS (IBM Corp., Version 25).

Considering the problem of multiple testing in statistics, we applied cluster based permutation tests implemented in the Fieldtrip toolbox *Maris and Oostenveld, 2007* to test for potential differences of EEG time-frequency data. Here, dependent sample *t*-tests and an independent sample test (for between subject group comparison of sleep data) were conducted. We specified the number of permutations by 1000 and the minimal number of channels by two. The cluster alpha level was set at p=0.05 for two-tailed testing. Plots with significant clusters of electrodes are shown in *Figures 2d and 4d*. After confirming the existence of a significant cluster, we conducted an additional post-hoc repeated-measure ANOVA with averaged values of the identified time and frequency range of interest and merged over the Pz and Cz electrodes (see *Figure 2e*). To control for SW amplitude size as a potential confounder in spindle activity differences during SW up-states, we extracted and averaged power values of the significant cluster (0.3–0.8 s; 11–14 Hz; Pz, P3, P4, O2, P7) per subject and conducted a post-hoc independent sample *t*-test based on residuals corrected by individual SW amplitude size. By using the same extracted power values (0.3–0.8 s; 11–14 Hz; Pz, P3, P4, O2, P7) per subject, we performed whole sample (n=22) Pearson correlation analyses between these power values and the overnight changes of behavior measurements of the cued condition (Δ d' and Δ c-criterion).

After conducting a sleep spindle detection (frequency range of 12–16 Hz, see Methods for details), we compared the sleep spindle density between the TMR conditions of high- and low-PP showing no significant difference (see *Figure 4—figure supplement 2a* and *Supplementary file 6*). Next, we subdivided the detected sleep spindles into coupled and uncoupled sleep spindles with the previously detected slow waves (SW; analyses of *Figure 4*). Sleep spindles were defined as coupled when their amplitude peak occurred during the SW up-state phase (0.3–0.8 s time-locked to the SW troughs). A two-way mixed design ANOVA on the amplitude size of the sleep spindles with the cueing group as a between-subject factor (high-PP-cued vs. low-PP-cued) and SW-coupling as a within-subject factor (coupled vs. uncoupled) showed a significant interaction effect (cueing group ×SW-coupling: $F_{(1,20)}$ = 4.51, p=0.046, $\eta^2$=0.18), a significant main effect of SW-coupling ($F_{(1,20)}$ = 85.02, p<0.001, $\eta^2$=0.81), and a trend of significance of the main effect of the cueing group ($F_{(1,20)}$ = 3.54, p=0.08). Post-hoc unpaired *t*-tests revealed a significant higher amplitude size of the coupled sleep spindles of the cueing group of high- compared to low-PP ($t_{(20)}$ = 2.13, p=0.046, Cohen's d=0.91; *Figure 4—figure supplement 2b*) and no significant group difference of the uncoupled sleep spindles ($t_{(20)}$ = 1.62, p=0.12). An additional comparison of the amount of coupled sleep spindles between the cueing groups revealed no significant difference (see *Supplementary file 6*).

Here, we found that detected sleep spindles coupled to the SW up-state phase occurred with higher amplitude after TMR presentations of the high-PP words in comparison to the low-PP words, whereas the sleep spindle density and the amount of sleep spindles coupled to the SW up-state phase did not differed between the cueing conditions.

## Acknowledgements

We thank the students at the University of Fribourg who assisted in data collection and Dr. Jonas Beck for reading and providing comments to improve the manuscript. This work was supported by the Swiss National Science Foundation Grant 168602 to A-LK and the European Research Council (ERC) under the European Union's Horizon 2020 research and innovation program (grant agreement MemoSleep No. 677875) to BR, and the University of Fribourg.

# Additional information

## Funding

| Funder | Grant reference number | Author |
|---|---|---|
| Schweizerischer Nationalfonds zur Förderung der Wissenschaftlichen Forschung | 168602 | Arndt-Lukas Klaassen |
| European Research Council | 677875 | Björn Rasch |

The funders had no role in study design, data collection and interpretation, or the decision to submit the work for publication.

## Author contributions

Arndt-Lukas Klaassen, Conceptualization, Data curation, Formal analysis, Funding acquisition, Investigation, Methodology, Project administration, Software, Validation, Visualization, Writing – original draft, Writing – review and editing; Björn Rasch, Conceptualization, Formal analysis, Funding acquisition, Investigation, Methodology, Project administration, Resources, Supervision, Validation, Visualization, Writing – original draft, Writing – review and editing

## Author ORCIDs

Arndt-Lukas Klaassen ⓘ https://orcid.org/0009-0009-8404-6295
Björn Rasch ⓘ https://orcid.org/0000-0001-7607-3415

## Ethics

The Ethical Commission of the Department of Psychology of the University of Fribourg approved this study (Ref-No.: 258). Informed consent was obtained from all participants.

Reviewer #1 (Public Review): https://doi.org/10.7554/eLife.90930.3.sa1
Reviewer #2 (Public Review): https://doi.org/10.7554/eLife.90930.3.sa2
Reviewer #3 (Public Review): https://doi.org/10.7554/eLife.90930.3.sa3
Author response https://doi.org/10.7554/eLife.90930.3.sa4

# Additional files

## Supplementary files

- Supplementary file 1. List of the high-PP words and their phoneme and biphone probabilities.

- Supplementary file 2. List of the low-PP words and their phoneme and biphone probabilities.

- Supplementary file 3. Gender, age and performance rates. Data are means ± SEM. PP, phonotactic probability. *P*-values of statistical comparisons between groups by using unpaired *t*-tests. Note, no significant group differences, but a trend of significance for pre-sleep memory test of c-criterion values of the high-PP condition.

- Supplementary file 4. Statistics of SW density time-bin analyses from –0.5–3 s. Data are means ± SEM. *P*-values (uncorrected for multiple comparisons) of statistical comparisons between groups by using paired *t*-tests against 0.

- Supplementary file 5. Sleep and reactivation parameter. Data are means ± SEM. N1, N2: Non-REM sleep stages N1, N2 and N3; REM, rapid-eye movement sleep; WASO, wake after sleep onset; SW, slow wave; PP, phonotactic probability. *P*-values of statistical comparisons between groups by using unpaired *t*-tests. Note, no significant group differences, but a trend of significance for REM sleep parameters.

- Supplementary file 6. Density parameters of detected sleep spindles. Data are means ± SEM. PP, phonotactic probability; SW, slow wave. General density: sleep spindle density during the time window of 0–6 s after stimulus onset in number per trial. SW-coupled: sleep spindles coupled to SW up-states divided by the total number of detected sleep spindles during the time window of interest

(0–6 s after stimulus onset) in percent. *P*-values of statistical comparisons between groups by using unpaired *t*-tests.

• MDAR checklist

## Data availability

All data and code used in the production of the figures and tables in this manuscript is freely available at https://osf.io/tjb9p/.

The following dataset was generated:

| Author(s) | Year | Dataset title | Dataset URL | Database and Identifier |
|---|---|---|---|---|
| Klaassen A-L, Rasch B | 2024 | TMR_Word_Learning_ Difficulty | https://osf.io/tjb9p/ | Open Science Framework, tjb9p |

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
