## [Editor Report · eLife assessment]

This study provides **useful** findings on how phonetic properties of words, i.e., their difficulty and prior knowledge, influence the outcome of targeted memory reactivation (TMR) during sleep. While these findings are supported by **solid** evidence, they are based on a small sample size warranting future work to shed further light on the impact of TMR in language learning.

---

## [Referee Report · Reviewer #1 (Public Review)]

Summary:

The authors aim to consider the effects of phonotactics on the effectiveness of memory reactivation during sleep. They have created artificial words that are either typical or atypical and showed that reactivation improves memory for the latter but not the former.

Strengths:

This is an interesting design and a creative way of manipulating memory strength and typicality. In addition, the spectral analysis on both the wakefulness data and the sleep data is well done. The article is clearly written and provides a relevant and comprehensive of the literature and of how the results contribute to it.

Weaknesses:

(1) Unlike most research involving artificial language or language in general, the task engaged in this manuscript did not require (or test) learning of meaning or translation. Instead, the artificial words were arbitrarily categorised and memory was tested for that categorisation. This somewhat limits the interpretation of the results as they pertain to language science, and qualifies comparisons with other language-related sleep studies that the manuscript builds on.

(2) Participants had to determine whether words are linked with reward or omission of punishment (if correctly categorised). Therefore, the task isn't a mere item categorisation task (group A/B), but also involves the complicated effects of reward (e.g., reward/loss asymmetries as predicted by prospect theory). This is not, in itself, a flaw, but there isn't a clear hypothesis as to the effects of reward on categorisation, and therefore no real justification for this design. This aspect of the task may add unneeded complexity (at best) or some reward-related contamination of the results (at worst).

(3) The study starts off with a sample size of N=39 but excludes 17 participants for some crucial analyses. This is a high number, and exclusion criteria were not pre-registered. Having said that, some criteria seem very reasonable (e.g., excluding participants who were not fully exposed to words during sleep).

(4) Relatedly, the final N is low for a between-subjects study (N=11 per group). This is adequately mentioned as a limitation, but since it does qualify the results, it seemed important to mention it here.

(5) The linguistic statistics used for establishing the artificial words are all based on American English, and are therefore in misalignment with the spoken language of the participants (which was German). This is a limitation of the study. The experimenters did not check whether participants were fluent in English. In all fairness, the behavioural effects presented in Figure 2A are convincing, providing a valuable manipulation test.

---

## [Referee Report · Reviewer #2 (Public Review)]

Summary:

The work by Klaassen & Rasch investigates the influence of word learning difficulty on sleep-associated consolidation and reactivation. They elicited reactivation during sleep by applying targeted memory reactivation (TMR) and manipulated word learning difficulty by creating words more similar (easy) or more dissimilar (difficult) to our language.

In one group of participants, they applied TMR of easy words and in another group of participants, they applied TMR of difficult words (between-subjects design).

They showed that TMR leads to higher memory benefits in the easy compared to the difficult word group. On a neural level, they showed an increase in spindle power (in the up-state of an evoked response) when easy words were presented during sleep.

Strengths:

The authors investigate a research question relevant to the field, that is, which experiences are actually consolidated during sleep. To address this question, they developed an innovative task and manipulated difficulty in an elegant way.

Overall, the paper is clearly structured, results and methods are described in an understandable way. The analyses approach is solid.

Weaknesses:

(1) Sample size

For a between-subjects design, the sample size is too small (N = 22). The main finding (also found in the title "Difficulty in artificial word learning impacts targeted memory reactivation") is based on an independent samples t-test with 11 participants/group.

The authors explicitly mention the small sample size and the between-subjects design as a limitation in their discussion. Nevertheless, making meaningful inferences based on studies with such a small sample size is difficult.

(2) Choice of task

Even though the task itself is innovative, there would have been tasks better suited to address the research question. The main disadvantage the task and the operationalisation of memory performance (d') have is that single-trial performance cannot be calculated. Consequently, choosing individual items for TMR is not possible.

Additionally, TMR of low vs. high difficulty is conducted between subjects (and independently of pre-sleep memory performance) which is a consequence of the task design.

---

## [Referee Report · Reviewer #3 (Public Review)]

Summary:

In this study, the authors investigated the effects of targeted memory reactivation (TMR) during sleep on memory retention for artificial words with varying levels of phonotactical similarity to real words. The authors report that the high phonotactic probability (PP) words showed a more pronounced EEG alpha decrease during encoding and were more easily learned than the low PP words. Following TMR during sleep, participants who had been cued with the high PP TMR, remembered those words better than 0, whilst no such difference was found in the other conditions. Accordingly, the authors report higher EEG spindle band power during slow-wave up-states for the high PP as compared to low PP TMR trials. Overall, the authors conclude that artificial words which are easier to learn benefit more from TMR than those which are difficult to learn.

Strengths:

(1) The authors have carefully designed the artificial stimuli to investigate the effectiveness of TMR on words that are easy to learn and difficult to learn due to their levels of similarity with prior word-sound knowledge. Their approach of varying the level of phonotactic probability enables them to have better control over phonotactical familiarity than in a natural language and are thus able to disentangle which properties of word learning contribute to TMR success.

(2) The use of EEG during wakeful encoding and sleep TMR sheds new light on the neural correlates of high PP vs low PP both during wakeful encoding and cue-induced retrieval during sleep.

Weaknesses:

(1) The present analyses are based on a small sample and comparisons between participants rather than within participants. Considering that the TMR benefits are based on changes in memory categorization between participants, it could be argued that the individuals in the high PP group were more susceptible to TMR than those in the low PP group for reasons other than the phonotactic probabilities of the stimuli (e.g., these individuals might be more attentive to sounds in the environment during sleep). While the authors acknowledge the small sample size and between-subjects comparison as a limitation, these results should be interpreted with caution.

Impact:

This work is likely to contribute to the subfield of sleep and memory, and their experimental methods could provide a useful resource for those which investigate memory processing of linguistic material.

---

## [Author Response]

The following is the authors’ response to the original reviews.

Public Reviews:Reviewer #1 (Public Review):Summary:The authors aim to consider the effects of phonotactics on the effectiveness of memory reactivation during sleep. They have created artificial words that are either typical or atypical and showed that reactivation improves memory for the latter but not the former.Comment 1:Strengths:This is an interesting design and a creative way of manipulating memory strength and typicality. In addition, the spectral analysis on both the wakefulness data and the sleep data is well done. The article is clearly written and provides a relevant and comprehensive of the literature and of how the results contribute to it.

We thank the reviewer for his/her positive evaluation of our manuscript.

Comment 2:Weaknesses:(1) Unlike most research involving artificial language or language in general, the task engaged in this manuscript did not require (or test) learning of meaning or translation. Instead, the artificial words were arbitrarily categorised and memory was tested for that categorisation. This somewhat limits the interpretation of the results as they pertain to language science, and qualifies comparisons with other language-related sleep studies that the manuscript builds on.

We thank the reviewer for this comment. We agree that we did not test for meaning or translation but used a categorization task in which we trained subjects to discriminate artificial words according to their reward associations (rewarded vs. non-rewarded). Previous language studies (Batterink et al., 2014; Batterink and Paller, 2017; Reber, 1967) used artificial words to investigate implicit learning of hidden grammar rules. Here, the language researchers studied generalization of the previously learned grammar knowledge by testing subject’s ability to categorize correctly a novel set of artificial words into rule-congruent versus rule-incongruent words. These differences to our study design might limit the comparability between the results of previous language studies of artificial grammar learning and our findings. We discussed now this aspect as a limitation of our novel paradigm.

We added the following sentences to the discussion on p.14, ll. 481-488:

Based on our paradigm, we investigated categorization learning of artificial words according to their reward associations (rewarded vs. unrewarded) and did not studied aspects of generalization learning of artificial grammar rules (Batterink et al., 2014; Batterink and Paller, 2017; Reber, 1967). This difference might limit the comparability between these previous language-related studies and our findings. However, the usage of artificial words with distinct phonotactical properties provided a successful way to manipulate learning difficulty and to investigate word properties on TMR, whereas our reward categorization learning paradigm had the advantage to increase the relevance of the word learnings due to incentives.

Comment 3:(2) The details of the behavioural task are hard to understand as described in the manuscript. Specifically, I wasn't able to understand when words were to be responded to with the left or right button. What were the instructions? Were half of the words randomly paired with left and half with right and then half of each rewarded and half unrewarded? Or was the task to know if a word was rewarded or not and right/left responses reflected the participants' guesses as to the reward (yes/no)? Please explain this fully in the methods, but also briefly in the caption to Figure 1 (e.g., panel C) and in the Results section.

We thank the reviewer for this comment and added additional sentences into the document to provide additional explanations. We instructed the participants to respond to each word by left- and right-hand button presses, whereas one button means the word is rewarded and the other button means the word is unrewarded. The assignment of left- and right-hand button presses to their meanings (rewarded versus unrewarded) differed across subjects. In the beginning, they had to guess. Then over trial repetitions with feedback at the end of each trial, they learned to respond correctly according to the rewarded/unrewarded associations of the words.

We added the following sentences to the results section on p.5, ll. 161-168:

As a two alternative forced-choice task, we assigned left- and right-hand button presses to the rewarded and the unrewarded word category, counterbalanced across subjects. We instructed the participants to respond to each word by left- or right-hand button presses, whereas one button means the word is rewarded (gain of money points) and the other button means the word is unrewarded (avoid the loss of money points). In the beginning, they had to guess. By three presentations of each word in randomized order and by feedback at the end of each trial, they learned to respond correctly according to the rewarded/unrewarded associations of the words (Fig. 1c).

We added the following sentences to the caption of Figure 1 on p.6, ll. 188-194:

As a two alternative forced-choice task, responses of left- and right-hand button presses were assigned to the rewarded and the unrewarded word category, respectively. The participants were instructed to respond to each word by left- or right-hand button presses, whereas one button means the word is rewarded (gain of money points) and the other button means the word is unrewarded (avoid the loss of money points). (d) Feedback matrix with the four answer types (hits: rewarded and correct; CR, correct rejections: unrewarded and correct; misses: rewarded and incorrect; FA, false alarms: unrewarded and incorrect) regarding to response and reward assignment of the word.

We added the following sentences to the methods on p.19, ll. 687-692:

As a two alternative forced-choice task, we assigned left- and right-hand button presses to the rewarded and the unrewarded word category, counterbalanced across subjects. We instructed the participants to respond to each word by left- or right-hand button presses, whereas one button means the word is rewarded (gain of money points) and the other button means the word is unrewarded (avoid the loss of money points).

Comment 4:(3) Relatedly, it is unclear how reward or lack thereof would translate cleanly into a categorisation of hits/misses/correct rejections/false alarms, as explained in the text and shown in Figure 1D. If the item was of the non-rewarded class and the participant got it correct, they avoided loss. Why would that be considered a correct rejection, as the text suggests? It is no less of a hit than the rewarded-correct, it's just the trial was set up in a way that limits gains. This seems to mix together signal detection nomenclature (in which reward is uniform and there are two options, one of which is correct and one isn't) and loss-aversion types of studies (in which reward is different for two types of stimuli, but for each type you can have H/M/CR/FA separably). Again, it might all stem from me not understanding the task, but at the very least this required extended explanations. Once the authors address this, they should also update Fig 1D. This complexity makes the results relatively hard to interpret and the merit of the manuscript hard to access. Unless there are strong hypotheses about reward's impact on memory (which, as far as I can see, are not at the core of the paper), there should be no difference in the manner in which the currently labelled "hits" and "CR" are deemed - both are correct memories. Treating them differently may have implications on the d', which is the main memory measure in the paper, and possibly on measures of decision bias that are used as well.

We thank the reviewer for this comment giving us the opportunity to clarify. As explained in the previous comment, for our two alternative forced-choice task, we instructed the participants to press one button when they were thinking the presented word is rewarded and the other button, when they were thinking the word is unrewarded. Based on this instruction, we applied the signal detection theory (SDT), because the subjects had the task to detect when reward was present or to reject when reward was absent. Therefore, we considered correct responses of words of the rewarded category as hits and words of the unrewarded category as correct rejections (see Table below). However, the reviewer is correct because in addition to false alarms, we punished here the incorrect responses by subtraction of money points to control for alternative task strategies of the participants instead of reward association learning of words. We agree that further explanation/argumentation to introduce our nomenclature is necessary.

**Author response table 1. sa4table1:** 

	Respond: Reward Present	Respond: Reward Absent
Reward Present	Hit	Miss
Reward Absent	False Alarm	Correct Rejection

We adjusted the results section on p.5, ll. 169-177:

To obtain a measurement of discrimination memory with respect to the potential influence of the response bias, we applied the signal detection theory (Green and Swets, 1966). Because, we instructed the participants to respond to each word by left- or right-hand button presses and that one button means reward is present whereas the other button means reward is absent, we considered correct responses of words of the rewarded category as hits and words of the unrewarded category as correct rejections. Accordingly, we assigned the responses with regard to the reward associations of the words to the following four response types: hits (rewarded, correct); correct rejections (unrewarded, correct); misses (rewarded, incorrect); and false alarms (unrewarded, incorrect). Dependent on responses, subjects received money points (Fig. 1d).

Comment 5:(4) The study starts off with a sample size of N=39 but excludes 17 participants for some crucial analyses. This is a high number, and it's not entirely clear from the text whether exclusion criteria were pre-registered or decided upon before looking at the data. Having said that, some criteria seem very reasonable (e.g., excluding participants who were not fully exposed to words during sleep). It would still be helpful to see that the trend remains when including all participants who had sufficient exposure during sleep. Also, please carefully mention for each analysis what the N was.

Our study was not pre-registered. Including all the subjects independent of low prememory performance, but with respect to a decent number of reactivations (> 160 reactivations, every word at least 2 times), resulted in a new dataset with 15 and 13 participants of the high- and low-PP cueing condition, respectively. Here, statistical analyses revealed no significant overnight change anymore in memory performance in the high-PP cueing condition (Δ memory (d'): t(14) = 1.67, p = 0.12), whereas the increase of the bias in decision making towards risk avoidance still remained significant (Δ bias (c-criterion): t(14) = 3.36, p = 0.005).

We modified and added the following sentences to the discussion on p.13, ll. 456-458:

Our study has limitations due to a small sample size and between-subject comparisons. The criteria of data analyses were not pre-registered and the p-values of our behavior analyses were not corrected for multiple comparisons.

Comment 6:(5) Relatedly, the final N is low for a between-subjects study (N=11 per group). This is adequately mentioned as a limitation, but since it does qualify the results, it seemed important to mention it in the public review.

We agree with the reviewer that the small sample size and the between subject comparisons represent major limitations of our study. Accordingly, we now discussed these limitations in more detail by adding alternative explanations and further suggestions for future research to overcome these limitations.

We added the following sentences to the discussion about the limitations on p.14, ll. 465-488:

To control for potential confounders despite the influence of difficulty in word learning on TMR, we compared parameters of sleep, the pre-sleep memory performance and the vigilance shortly before the post-sleep memory test, revealing no significant group differences (see Table S1 and S2). Nevertheless, we cannot rule out that other individual trait factors differed between the groups, such as the individual susceptibility to TMR. To rule out these alternative explanations based on individual factors, we suggest for future research to replicate our study by conducting a within-subject design with cueing of subsets of previously learned low- and high-PP words providing all conditions within the same individuals as shown in other TMR studies (Cairney et al., 2018; Schreiner and Rasch, 2015).

Comment 7:(6) The linguistic statistics used for establishing the artificial words are all based on American English, and are therefore in misalignment with the spoken language of the participants (which was German). The authors should address this limitation and discuss possible differences between the languages. Also, if the authors checked whether participants were fluent in English they should report these results and possibly consider them in their analyses. In all fairness, the behavioural effects presented in Figure 2A are convincing, providing a valuable manipulation test.

We thank the reviewer pointing to the misalignment between the German-speaking participants and the used artificial words based on American English. Further, we did not assessed the English language capability of the participants to control it as a potential confounder, whereas comparative control analyses revealed no significant differences between the both cueing groups in pre-sleep memory performance (see Table S1).

We now discussed these comments as limitations on p.14, ll. 473-481:

Further, we used artificial words based on American English in combination with German speaking participants, whereas language differences of pronunciation and phoneme structures might affect word perception and memory processing (Bohn and Best, 2012). On the other hand, both languages are considered to have the same language family (Eberhard et al., 2019) and the phonological distance between English and German is quite short compared for example to Korean (Luef and Resnik, 2023). Thus, major common phonological characteristics across both languages are still preserved. In addition, our behavior analyses revealed robust word discrimination learning and distinct memory performance according to different levels of phonotactic probabilities providing evidence of successful experimental manipulation.

Comment 8:(7) With regard to the higher probability of nested spindles for the high- vs low-PP cueing conditions, the authors should try and explore whether what the results show is a general increase for spindles altogether (as has been reported in the past to be correlated with TMR benefit and sleep more generally) or a specific increase in nested spindles (with no significant change in the absolute numbers of post-cue spindles). In both cases, the results would be interesting, but differentiating the two is necessary in order to make the claim that nesting is what increased rather than spindle density altogether, regardless of the SW phase.

We conducted additional analyses based on detected sleep spindles to provide additional data according to this question.

We added the following section to the supplementary data on pp. 31-32, ll. 1007-1045:

After conducting a sleep spindle detection (frequency range of 12-16Hz, see methods for details), we compared the sleep spindle density between the TMR conditions of high- and lowPP showing no significant difference (see Fig. S8a and Table S9). Next, we subdivided the detected sleep spindles into coupled and uncoupled sleep spindles with the previously detected slow waves (SW; analyses of Fig. 4). Sleep spindles were defined as coupled when their amplitude peak occurred during the SW up-state phase (0.3 to 0.8s time-locked to the SW troughs). A two-way mixed design ANOVA on the amplitude size of the sleep spindles with the cueing group as a between-subject factor (high-PP-cued vs. low-PP-cued) and SW-coupling as a within-subject factor (coupled vs. uncoupled) showed a significant interaction effect (cueing group × SW-coupling: F(1,20) = 4.51, p = 0.046, η2 = 0.18), a significant main effect of SW-coupling (F(1,20) = 85.02, p < 0.001, η2 = 0.81), and a trend of significance of the main effect of the cueing group (F(1,20) = 3.54, p = 0.08). Post-hoc unpaired t-tests revealed a significant higher amplitude size of the coupled sleep spindles of the cueing group of high- compared to low-PP (t(20) = 2.13, p = 0.046, Cohen’s d = 0.91; Fig. S8b) and no significant group difference of the uncoupled sleep spindles (t(20) = 1.62, p = 0.12). An additional comparison of the amount of coupled sleep spindles between the cueing groups revealed no significant difference (see Table S9).

Here, we found that detected sleep spindles coupled to the SW up-state phase occurred with higher amplitude after TMR presentations of the high-PP words in comparison to the low-PP words, whereas the sleep spindle density and the amount of sleep spindles coupled to the SW up-state phase did not differed between the cueing conditions.

We added the following sentences to the methods on pp. 22-23, ll. 822-839:

Sleep spindle analyses

We detected fast sleep spindles by band-pass filtering (12-16Hz) the signal of the Pz electrode during the auditory cueing trials in the time windows of -2 to 8s according to stimulus onsets. The amplitude threshold was calculated individually for each subject as 1.25 standard deviations (SDs) from the mean. The beginning and end times of the sleep spindles were then defined as the points at which the amplitude fell below 0.75 SDs before and after the detected sleep spindle. Only sleep spindles with a duration of 0.5-3 s were included in subsequent analyses.

To compare the sleep spindle densities between the different cueing conditions of high- and low-PP, we computed the grand average sleep spindle density distribution in number per trial with a bin size of 0.5s from -0.5 to 6s time-locked to stimulus onset in each condition (see Fig. S8a and Table S9).

Based on the detected slow waves and sleep spindles, we defined coupling events when the positive amplitude peak of a detected sleep spindle was occurring during the slow wave upstate phase in a time window of 0.3 to 0.8s according to the trough of a slow wave.

We computed the averaged amplitude size of each detected sleep spindle by calculating the mean of the absolute amplitude values of all negative and positive peaks within a detected sleep spindle (see Fig. S8b).

We added the following sentences to the results on p.10, ll. 338-343:

By conducting an additional analyses based on detection of fast sleep spindles (12-16Hz; see methods), we confirmed that fast sleep spindles during the SW up-states (from 0.3 to 0.8s after the SW trough) occurred with significantly higher amplitude after the cueing presentation of high- compared to low-PP words, whereas parameters of sleep spindle density and the amount sleep spindles coupled to the SW up-state did not differed between the cueing conditions (see Fig. S8 and Table S9).

**Reviewer #2 (Public Review):**
Summary:The work by Klaassen & Rasch investigates the influence of word learning difficulty on sleepassociated consolidation and reactivation. They elicited reactivation during sleep by applying targeted memory reactivation (TMR) and manipulated word learning difficulty by creating words more similar (easy) or more dissimilar (difficult) to our language. In one group of participants, they applied TMR of easy words and in another group of participants, they applied TMR of difficult words (between-subjects design). They showed that TMR leads to higher memory benefits in the easy compared to the difficult word group. On a neural level, they showed an increase in spindle power (in the up-state of an evoked response) when easy words were presented during sleep.Comment 9:Strengths:The authors investigate a research question relevant to the field, that is, which experiences are actually consolidated during sleep. To address this question, they developed an innovative task and manipulated difficulty in an elegant way.Overall, the paper is clearly structured, and results and methods are described in an understandable way. The analysis approach is solid.

We thank the reviewer for his/her positive evaluation of our manuscript.

Weaknesses:Comment 10:(1) Sample sizeFor a between-subjects design, the sample size is too small (N = 22). The main finding (also found in the title "Difficulty in artificial word learning impacts targeted memory reactivation") is based on an independent samples t-test with 11 participants/group.The authors explicitly mention the small sample size and the between-subjects design as a limitation in their discussion. Nevertheless, making meaningful inferences based on studies with such a small sample size is difficult, if not impossible.

We agree with the reviewer that the small sample size and the between subject comparisons represent major limitations of our study. Accordingly, we now discussed these limitations in more detail by adding alternative explanations and further suggestions for future research to overcome these limitations.

We added the following sentences to the discussion about the limitations on p.14, ll. 465-473:

To control for potential confounders despite the influence of difficulty in word learning on TMR, we compared parameters of sleep, the pre-sleep memory performance and the vigilance shortly before the post-sleep memory test, revealing no significant group differences (see TableS1 and S2). Nevertheless, we cannot rule out that other individual trait factors differed between the groups, such as the individual susceptibility to TMR. To rule out these alternative explanations based on individual factors, we suggest for future research to replicate our study by conducting a within-subject design with cueing of subsets of previously learned low- and high-PP words providing all conditions within the same individuals as shown in other TMR studies (Cairney et al., 2018; Schreiner and Rasch, 2015).

Comment 11:(2) Choice of taskthough the task itself is innovative, there would have been tasks better suited to address the research question. The main disadvantage the task and the operationalisation of memory performance (d') have is that single-trial performance cannot be calculated. Consequently, choosing individual items for TMR is not possible.Additionally, TMR of low vs. high difficulty is conducted between subjects (and independently of pre-sleep memory performance) which is a consequence of the task design.The motivation for why this task has been used is missing in the paper.

We used a reward task combined with TMR because previous studies revealed beneficial effects of reward related information on sleep dependent memory consolidation and reactivation (Asfestani et al., 2020; Fischer and Born, 2009; Lansink et al., 2009; Sterpenich et al., 2021). In addition, we wanted to increase the motivation of the participants, as they could receive additional monetary compensation according to their learning and memory task performances. Furthermore, we designed the task, with the overall possibility to translate this task to operant conditioning in rats (see research proposal: https://data.snf.ch/grants/grant/168602). However, the task turned out to be too difficult to translate to rats, whereas we developed a different learning paradigm for the animal study (Klaassen et al., 2021) of this cross-species research project.

We added the following sentence to the introduction on p.4, ll. 134-137:

To consider the beneficial effect of reward related information on sleep dependent memory consolidation and reactivation (Asfestani et al., 2020; Fischer and Born, 2009; Lansink et al., 2009; Sterpenich et al., 2021), we trained healthy young participants to categorize these words into rewarded and unrewarded words to gain and to avoid losses of money points.

**Reviewer #3 (Public Review):**
Summary:In this study, the authors investigated the effects of targeted memory reactivation (TMR) during sleep on memory retention for artificial words with varying levels of phonotactical similarity to real words. The authors report that the high phonotactic probability (PP) words showed a more pronounced EEG alpha decrease during encoding and were more easily learned than the low PP words. Following TMR during sleep, participants who had been cued with the high PP TMR, remembered those words better than 0, whilst no such difference was found in the other conditions. Accordingly, the authors report higher EEG spindle band power during slow-wave up-states for the high PP as compared to low PP TMR trials. Overall, the authors conclude that artificial words that are easier to learn, benefit more from TMR than those which are difficult to learn.Comment 12 & 13:Strengths:(1) The authors have carefully designed the artificial stimuli to investigate the effectiveness of TMR on words that are easy to learn and difficult to learn due to their levels of similarity with prior wordsound knowledge. Their approach of varying the level of phonotactic probability enables them to have better control over phonotactical familiarity than in a natural language and are thus able to disentangle which properties of word learning contribute to TMR success.(2) The use of EEG during wakeful encoding and sleep TMR sheds new light on the neural correlates of high PP vs. low PP both during wakeful encoding and cue-induced retrieval during sleep.

We thank the reviewer for his/her positive evaluation of our manuscript.

Weaknesses:Comment 14:(1) The present analyses are based on a small sample and comparisons between participants. Considering that the TMR benefits are based on changes in memory categorization between participants, it could be argued that the individuals in the high PP group were more susceptible to TMR than those in the low PP group for reasons other than the phonotactic probabilities of the stimuli (e.g., these individuals might be more attentive to sounds in the environment during sleep). While the authors acknowledge the small sample size and between-subjects comparison as a limitation, a discussion of an alternative interpretation of the data is missing.

We agree with the reviewer that the small sample size and the between subject comparisons represent major limitations of our study. We thank the reviewer for this helpful comment and now discussed these limitations in more detail by adding alternative explanations and further suggestions for future research to overcome these limitations.

We added the following sentences to the discussion on p.14, ll. 465-473:

To control for potential confounders despite the influence of difficulty in word learning on TMR, we compared parameters of sleep, the pre-sleep memory performance and the vigilance shortly before the post-sleep memory test, revealing no significant group differences (see Table S1 and S2). Nevertheless, we cannot rule out that other individual trait factors differed between the groups, such as the individual susceptibility to TMR. To rule out these alternative explanations based on individual factors, we suggest for future research to replicate our study by conducting a within-subject design with cueing of subsets of previously learned low- and high-PP words providing all conditions within the same individuals as shown in other TMR studies (Cairney et al., 2018; Schreiner and Rasch, 2015).

Comment 15:(2) While the one-tailed comparison between the high PP condition and 0 is significant, the ANOVA comparing the four conditions (between subjects: cued/non-cued, within-subjects: high/low PP) does not show a significant effect. With a non-significant interaction, I would consider it statistically inappropriate to conduct post-hoc tests comparing the conditions against each other. Furthermore, it is unclear whether the p-values reported for the t-tests have been corrected for multiple comparisons. Thus, these findings should be interpreted with caution.

We thank the reviewer for this comment giving us the opportunity to correct our analyses and clarify with additional description. Indeed, we investigated at first overnight changes in behavior performance within the four conditions, conducting t-tests against 0 of Δ-values of d' and c-criterion. Whereas for all our statistical analyses the p-value was set at p < 0.05 for two-tailed testing, we did not corrected the p-value of our behavior analyses for multiple comparisons. To investigate subsequently differences between conditions, we conducted additional ANOVAs. We agree with the reviewer that without significant of results of the ANOVA, post-hoc analyses should not be conducted. Taken in account as well the recommendation of reviewer 1, we included now only post-hoc pairwise comparisons when the interaction effect of the ANOVA revealed at least a trend of significance (p < 0.1).

We removed the following post-hoc analyses from the results section on p.9, ll. 291-295:

Additional post-hoc pairwise comparisons revealed a significant difference between the highPP cued and low-PP uncued (high-PP cued vs. low-PP uncued: t(10) = 2.43, p = 0.04), and no difference to other conditions (high-PP cued vs.: high-PP uncued t(20) = 1.28, p = 0.22; lowPP cued t(20) = 1.57, p = 0.13).

Further, we mentioned the lack of correction for multiple comparisons as a limitation of our results in the discussion on p.13, ll. 456-458:

The criteria of data analyses were not pre-registered and the p-values of our behavior analyses were not corrected for multiple comparisons.

We added the following sentences to the methods p.23, ll. 842-849:

To analyze overnight changes of sleep behavioral data within TMR conditions, we conducted at first dependent sample t-tests against 0 of Δ-values (post-sleep test minus pre-sleep test) of d' and c-criterion (see Fig. 3). Two-way mixed design ANOVAs were computed to compare Δvalues between TMR conditions. After confirming at least a trend of significance (p < 0.1) for the interaction effect, we conducted post-hoc pairwise comparisons by independent and dependent sample t-tests. For all behavior statistical analyses, the p-value was set at p < 0.05 for two-tailed testing. A p-value < 0.1 and > 0.05 was reported as a trend of significance.

Comment 16:(3) With the assumption that the artificial words in the study have different levels of phonotactic similarity to prior word-sound knowledge, it was surprising to find that the phonotactic probabilities were calculated based on an American English lexicon whilst the participants were German speakers. While it may be the case that the between-language lexicons overlap, it would be reassuring to see some evidence of this, as the level of phonotactic probability is a key manipulation in the study.

We thank the reviewer pointing to the misalignment between the German-speaking participants and the used artificial words based on American English. In line with this recommendation, we added a more outlined argumentation to the manuscript about the assumption of our study that major common phonetic characteristics across both languages are still preserved.

We now discussed these aspects on p.14, ll. 473-481:

Further, we used artificial words based on American English in combination with German speaking participants, whereas language differences of pronunciation and phoneme structures might affect word perception and memory processing (Bohn and Best, 2012). On the other hand, both languages are considered to have the same language family (Eberhard et al., 2019) and the phonological distance between English and German is quite short compared for example to Korean (Luef and Resnik, 2023). Thus, major common phonological characteristics across both languages are still preserved. In addition, our behavior analyses revealed robust word discrimination learning and distinct memory performance according to different levels of phonotactic probabilities providing evidence of successful experimental manipulation.

Comment 17:(4) Another manipulation in the study is that participants learn whether the words are linked to a monetary reward or not, however, the rationale for this manipulation is unclear. For instance, it is unclear whether the authors expect the reward to interact with the TMR effects.

We used a reward task combined with TMR because previous studies revealed beneficial effects of reward related information on sleep dependent memory consolidation and reactivation (Asfestani et al., 2020; Fischer and Born, 2009; Lansink et al., 2009; Sterpenich et al., 2021). In addition, we wanted to increase the motivation of the participants, as they could receive additional monetary compensation according to their learning and memory task performances. Furthermore, we designed the task, with the overall possibility to translate this task to operant conditioning in rats (see research proposal: https://data.snf.ch/grants/grant/168602). However, the task turned out to be too difficult to translate to rats, whereas we developed a different learning paradigm for the animal study (Klaassen et al., 2021) of this cross-species research project.

We added the following sentence to the introduction on p.4, ll. 134-137:

To consider the beneficial effect of reward related information on sleep dependent memory consolidation and reactivation (Asfestani et al., 2020; Fischer and Born, 2009; Lansink et al., 2009; Sterpenich et al., 2021), we trained healthy young participants to categorize these words into rewarded and unrewarded words to gain and to avoid losses of money points.

**Recommendations for the authors:**

**Reviewer #1 (Recommendations For The Authors):**
Comment 18:(1) Please clearly define all linguistics terms - and most importantly the term "phonotactics" - at first use.

We thank the reviewer for this recommendation and we added the definition of phonotactics and further reduced the diversity of linguistic terms to improve readability.

We added the following sentences to the beginning of the introduction on p.3, ll. 72-76:

One critical characteristic of similarity to pre-existing knowledge in auditory word processing is its speech sound (phoneme) pattern. In phonology as the field of language specific phoneme structures, phonotactics determines the constraints of word phoneme composition of a specific language.

Comment 19:(2) Some critical details about the methods should be included in the Results section to make it comprehensible. For example, the way the crucial differences between G1-4 words should be addressed in the Results, not only in Figure 1.

According to the recommendation, we added this information to the results section. We added the following sentences to the results section on p.4, ll. 145-154:

To study the impact of difficulty in word learning on TMR, we developed a novel learning paradigm. We formed four sets of artificial words (40 words per set; see Table S3 and S4) consisting of different sequences of two vowels and two consonants. Here, we subdivided the alphabet into two groups of consonants (C1: b, c, d, f, g, h, j, k, l, m; C2: n, p, q, r, s, t, v, w, x, z) and vowels (V1: a, e, I; V2: o, u, y). Four-letter-words were created by selecting letters from the vowel and consonant groups according to four different sequences (G1:C1, V1, V2, C2; G2: C1, V1, C2, V2; G3: V1, C1, C2, V2; G4: V1, C1, V2, C2; Fig. 1a; see methods for further details). Comparison analyses between the sets revealed significant differences in phonotactic probability (PP; Fig. 1b; unpaired t-tests: G1 / G2 > G3 / G4, p < 0.005, values of Cohen’s d > 0.71).

Comment 20(3) Was scoring done both online and then verified offline? If so, please note that.

We included now this information.

We adjusted the method section on p.21, ll. 765-769:

The sleep stages of NREM 1 to 3 (N1 to N3), wake, and REM sleep were scored offline and manually according to the criteria of the American Academy of Sleep Medicine (AASM) by visual inspection of the signals of the frontal, central, and occipital electrodes over 30s epochs (Iber et al., 2007). Based on offline scoring, we confirmed TMR exposure during N2 and N3 and no significant differences (p-values > 0.05) of sleep parameters between the cueing groups (see Table S2).

Comment 21:(4) In Figure 2, please arrange the panel letters in an easier-to-read way (e.g., label upper right panel b with a different letter).

Now we rearranged the panel letters according to the recommendation.

We adjusted Figure 2 on p.8, ll. 242-258:

Comment 22(5) In the first paragraph on TMR effects, please note which memory measure you are comparing (i.e., d').

We added this information according to the recommendation.

We adjusted the sentence of the results on p.8, ll. 260-263:

To examine whether TMR during sleep impacts memory consolidation of discrimination learning with respect to learning difficulty, we calculated the overnight changes by subtracting the pre- from the post-sleep memory performance based on d'-values of the reactivated sequences (cued) and non-reactivated sequences (uncued).

Comment 23:(6) Please show the pre-sleep and post-sleep test scores for both word categories (not only the delta). It may be best to show this as another data point in Fig 2a, but it may be helpful to also see this split between cued and uncued.

We added the pre-sleep and post-sleep test scores with the individual data points as an additional figure.

We added the following figure to the supplementary data on p.28, ll. 936-940:

Comment 24:(7) In the sentence "An additional two-way mixed design ANOVA on the same values with cueing as a between-subject factor (cued vs. uncued) ...", a more exact phrasing for the last parentheses would probably be "(high-PP-Cued vs Low-PP-Cued)". Both groups were cued.

We thank the reviewer pointing this out. According to the recommendation, we corrected the descriptions of the two-way mixed design ANOVAs. In addition, we detected a mistake of wrong assignments of the conditions to ANOVAs and corrected the reported values.

We adjusted the sentences and corrected the values on p.9, ll. 271-275 and ll. 289-291:

An additional two-way mixed design ANOVA on the same values with the factor cueing (cued vs. uncued) as a within-subject factor and group as a between-subject factor revealed trends of significance (p < 0.1) for the interaction (cueing × group: F(1,20) = 3.47, p = 0.08) and the main effect of group (F(1,20) = 3.28, p = 0.09). The main effect of cueing was not significant (F(1,20) = 0.58, p = 0.46).

An ANOVA on c-criterion changes showed no significant effects (interaction cueing × group: F(1,20) = 2.66, p = 0.12; main effect cueing F(1,20) = 2.08, p = 0.17; main effect group F(1,20) = 0.38, p = 0.55).

Comment 25:(8) In the same ANOVA, please mention that there is a trend toward an interaction effect. If there wasn't one, the post-hoc comparison would be unwarranted. Please consider noting other p<0.1 pvalues as a trend as well, for consistency.

Regarding this recommendation, we included now only post-hoc pairwise comparisons after confirming at least a trend toward an interaction effect of these ANOVAs and reported consistently a p-value < 0.1 and > 0.05 as a trend of significance.

We added the following sentences to the methods p.23, ll. 844-849:

Two-way mixed design ANOVAs were computed to compare Δ-values between TMR conditions. After confirming at least a trend of significance (p < 0.1) for the interaction effect, we conducted post-hoc pairwise comparisons by independent and dependent sample t-tests. For all behavior statistical analyses, the p-value was set at p < 0.05 for two-tailed testing. A p-value < 0.1 and > 0.05 was reported as a trend of significance.

We removed the following post-hoc analyses from the results section on p.9, ll. 291-295:

Additional post-hoc pairwise comparisons revealed a significant difference between the highPP cued and low-PP uncued (high-PP cued vs. low-PP uncued: t(10) = 2.43, p = 0.04), and no difference to other conditions (high-PP cued vs.: high-PP uncued t(20) = 1.28, p = 0.22; lowPP cued t(20) = 1.57, p = 0.13).

Comment 26:(9) Please consider adding an analysis correlating spindle power with memory benefit across participants. Even if it is non-significant, it is important to report given that some studies have found such a relationship.

According to this recommendation, we conducted an additional correlation analyses.

We added the following sentences to the manuscript into the results (pp. 10-11, ll. 346-349), the discussion (p.12, ll. 413-417), and the methods (p.23, ll. 864-867):

Whereas we found a significant group difference in spindle power nested during SW up-states, conducting further whole sample (n = 22) correlation analyses between the individual spindle power values of the significant cluster and the overnight changes of behavior measurements revealed no significant correlations (Δ d': r = 0.16, p = 0.48; Δ c-criterion: r = 0.19, p = 0.40).

In addition to our result of the significant group difference, we failed to find significant correlations between SW nested spindle power values and overnight changes in behavior measurements, whereas previous studies reported associations of SW and spindle activities during sleep with the integration of new memories in pre-existing knowledge networks (Tamminen et al., 2013, 2010).

By using the same extracted power values (0.3 to 0.8s; 11-14Hz; Pz, P3, P4, O2, P7) per subject, we performed whole sample (n = 22) Pearson correlation analyses between these power values and the overnight changes of behavior measurements of the cued condition (Δ d' and Δ ccriterion).

**Reviewer #2 (Recommendations For The Authors):**
(1) Choice of taskComment 27:In general, I find your task well-designed and novel. In light of your research question, however, I wonder why you chose this task. When you outlined the research question in the introduction, I expected a task similar to Schreiner et al. (2015). For example, participants have to associate high PP words with each other and low PP words. The advantage here would be that you could test the benefits of TMR in a within-subjects design (for example, cueing half of the remembered high and half of the remembered low PP words).

Please see our previous response at comment 14.

Comment 28:Why did you decide to introduce a reward manipulation?

Please see our previous response at comment 11.

Comment 29:Why did you do the cueing on a category level (cueing all high PP or all low PP words instead of single word cueing or instead of cueing 20 reward high-PP, 20 unrewarded high-PP plus 20 reward low-PP and 20 unrewarded low-PP)? Both alternatives would have provided you the option to run your statistics within participants.

Please see our previous response at comment 14.

Comment 30:(2) Between-subjects design and small sample size.Why did you decide on a between-subjects design that severely reduces your power?Why did you just collect 22 participants with such a design? Were there any reasons for this small sample size? Honestly, I think publishing a TMR study with healthy participants and such a small sample size (11 participants for some comparisons) is not advisable.

Please see our previous response at comment 14.

Comment 31:(3) Encoding performance.Is d' significantly above 0 in the first repetition round? I would assume that the distinction between rewarded and non-rewarded words is just possible after the first round of feedback.

Indeed, conducting t-tests against 0 revealed significantly increased d'-values in the first repetition round (2nd presentation) in both PP conditions (high-PP: 0.85 ± 0.09, t(32) = 9.17, p < 0.001; low-PP: 0.62 ± 0.09, t(32) = 6.83, p < 0.001).

Comment 32:(4) Encoding response optionsIf you want to you could make it more explicit what exactly the response options are. I assume that one button means a word has a high reward and the other button means a word has a low reward. Making it explicit increases the understanding of the results section.

Please see our previous response at comment 3.

Comment 33:(5) Alpha desynchronisation.Relative changeWhy did you subtract alpha power during the 1st presentation from alpha power during 2nd and 3rd presentation? You baseline-corrected already and individually included the 1st, 2nd, and 3rd repetition in your behavioural analysis.

Based on this analysis, we aimed to examine the relative change in alpha power between PP-conditions of memory-relevant word repetitions. Therefore, to extract memory relevant changes of EEG activities, the first word presentation of naive stimulus processing could serve as a more representative baseline condition covering the time-window of interest of 0.7 to 1.9 s after the stimulus onset compared to a baseline condition before stimulus onset (-1 to -0.1s).

To explain the rational of the analyses with the baseline condition more clearly, we added this information to the results section on p.7, ll. 222-226:

We obtained the changes in power values by subtracting the first from the second and third presentation for the high- and low-PP condition, respectively. Here, the first word presentation of naive stimulus processing served us with a more representative baseline condition covering the time-window of interest of 0.7 to 1.9 s after the stimulus onset to examine relevant changes of encoding.

Comment 34:(6) Alpha desynchronisation as a neural correlate of encoding depth & difficulty?"In addition to the behavior results, these EEG results indicate differences between PP conditions in desynchronization of alpha oscillations, as an assumed neural correlate of encoding depth. In addition to the behavior results, these EEG results indicate differences between PP conditions in desynchronization of alpha oscillations, as an assumed neural correlate of encoding depth."Given that the low-PP words are more difficult to learn, I was expecting to see higher alpha desynchronisation in the low-PP relative to the high-PP words. Could you outline in a bit more detail how your findings fit into the literature (e.g., Simon Hanslmayr did a lot of work on this)?I would also advise you to add citations e.g., after your sentence in the quote above ("as an assumed neural correlate of encoding depth").

We thank the reviewer for the recommendation giving us the opportunity to discuss in more detail how our results relate to previous findings.

We added additional sentences to the discussion on p.13, ll. 441-455:

Additional studies linked alpha desynchronization to cognitive effort and cognitive load (Proskovec et al., 2019; Zhu et al., 2021). So, one could assume to observe higher alpha desynchronization in the more difficult to learn condition of low-PP compared to high-PP. On the other hand numerous studies investigating oscillatory correlates of learning and memory showed that alpha desynchronization is associated with memory across different tasks, modalities and experimental phases of encoding and retrieval (Griffiths et al., 2016, 2021, 2019a, 2019b; Hanslmayr et al., 2009; Michelmann et al., 2016). Strikingly, Griffith and colleagues (Griffiths et al., 2019a) revealed by simultaneous EEG-fMRI recordings a negative correlation between the occurrence of patterns of stimulus-specific information detected by fMRI and cortical alpha/beta suppression. Here, the authors suggested that a decrease of alpha/beta oscillations might represent the neuronal mechanism of unmasking the task-critical signal by simultaneous suppression of task-irrelevant neuronal activities to promote information processing. Following this interpretation, we assume that over the course of learning elevated memory processing of the easier to learn stimuli is associated with enhanced information processing and thus accompanied by higher cortical alpha desynchronization in comparison of the more difficult to learn stimuli.

In addition, we added the mentioned quote on p.7, ll. 239-240:

In addition to the behavior results, these EEG results indicate differences between PP conditions in desynchronization of alpha oscillations, as an assumed neural correlate of encoding depth (Griffiths et al., 2021; Hanslmayr et al., 2009).

Comment 35:(7) Exclusion criterion.Why did you use a d' > 0.9 as a criterion for data inclusion?

This criterion ensured that each included subject had at least in one PP-condition a d' > 1.05 of pre-sleep memory performance, which corresponds to a general accuracy rate of 70%.

Accordingly, we adjusted these sentences of the method section on p.19, ll. 677-680:

Data were excluded from subjects who did not reach the minimal learning performance of d' > 1.05 during the pre-sleep memory test in at least one of the two PP conditions, whereas this threshold value corresponds to accuracy rates of 70% (n = 5). In addition, we excluded one subject who showed a negative d' in one PP condition of the pre-sleep memory test (n = 1).

Comment 36:(8) Coherence of wording.When you talk about your dependent variable (d') you sometimes use sensitivity. I would stick to one term.

We replaced the word sensitivity with d'.

(9) CriterionComment 37:Why do you refer to a change in criterion (Figure 3b, axis labels) as a change in memory? Do you think the criterion says something about memory?

We corrected the axis label of Figure 3b and deleted here the word memory.

Comment 38:Additionally, why did you analyse the effect of TMR on the criterion? Do you expect the criterion to change due to sleep-dependent memory consolidation? This section would benefit from more explanation. Personally, I am very interested in your thoughts and your hypothesis (if you had one, if not that is also fine but then, make it explicit that it was an exploratory analysis).

By conducting exploratory analyses of overnight changes of the c-criterion measurements, we aimed to examine the bias of decision-making to provide comprehensive data according to the framework of the signal detection theory. Regarding the previous literature showing mainly beneficial effects of sleep on learning and memory, we focused with our hypothesis on d' and explored additionally the c-criterion.

Despite our task design with gains/hits of +10 money points and losses/FAs of -8 (instead of -10), the subjects showed already during the pre-sleep memory task significant biases towards loss avoidance in both PP conditions (t-tests against 0: high-PP: 0.44 ± 0.07, t(21) = 5.63, p < 0.001; low-PP: 0.47 ± 0.09, t(21) = 5.51, p < 0.001). As already reported in the preprint, we found an additional significant increase of c-criterion by TMR solely for the high-PP words (see Fig. 3b). Even by integrating subjects with poor pre-sleep memory performance (high-PP-cueing group: n = 15; low-PP-cueing group: n = 13), t-tests against 0 revealed a significant increase of the high-PP cueing condition (t(14) = 3.36, p = 0.005) and no significant overnight changes in the other conditions (high-PP uncued: t(12) = 1.39, p = 0.19; low-PP cued: t(12) = 1.47, p = 0.17; low-PP uncued: t(14) = -0.20, p = 0.84). These exploratory findings on c-criterion suggest potential applications of TMR to affect decision-making biases in combination with reward learning.

We revised the manuscript mentioning the exploratory character of the c-criterion analyses of the results on p.9, ll. 282-283 and of the discussion on p.12, ll. 400-402:

We examined next as an exploratory analysis whether TMR conditions influence biases in decision-making.

By conducting an additional exploratory analysis, we observed a significant change of the decision bias in the cueing condition of the easy to learn words and no overnight changes in the other conditions.

Comment 39:(10) You detected SWs in the time range of 0-6 sec post sound stimulation. How was the distribution of all detected SW down-states in this time range? (You could plot a histogram for this.)

We illustrated now the detected SWs in the time range of 0 to 6 s after stimulus onset.

We added a histogram to the supplementary section on p.30, ll. 982-986:

**Reviewer #3 (Recommendations For The Authors):**
Comment 40:(1) In line with the weakness outlined above, I would recommend including a discussion of how the between-subject comparison and small sample size could affect the results and provide alternative interpretations.

Please see our previous response at comment 14.

Comment 41:(2) Regarding my point about statistical comparisons, I would recommend that the authors follow best practice guidelines for post-hoc tests and multiple comparisons. In Figures 3a and b, I would also recommend removing the stars indicating significance from the post-hoc tests (if this is what they reflect). Perhaps this link will be useful: https://www.statology.org/anova-post-hoc-tests/

Please see our previous response at comment 15.

Comment 42:(3) Furthermore, to address any doubts about the possible phonotactic probability differences between languages, I would recommend that the authors show whether the languages overlap, the level of English fluency in the German-speaking participants, and/or another way of reassuring that this is unlikely to have affected the results.

Please see our previous response at comment 7.

Comment 43:(4) In the introduction, I would recommend that the authors outline a clear rationale for the reward/no reward manipulation.

Please see our previous response at comment 11.

Comment 44:(5) Figure 1c: Please include what response options participants had, e.g., 'rewarded/not rewarded'. This would make the type of categorization clearer to the reader.

Please see our previous response at comment 3.

Comment 45:(6) It is unclear whether the additional ANOVA conducted on the time and frequency of the identified clusters included all channels or only the channels contributing to the cluster. Consider clarifying this in the relevant methods and results. Furthermore, I would recommend labelling this as a posthoc test as this analysis was guided by an initial peak at the data and the timings, frequencies, and channels of interest were not selected a-priori.

We thank the reviewer for this recommendation and labelled the additional repeatedmeasure ANOVA as a post-hoc test. Further, we mentioned the used channels (Pz and Cz) for this analyses.

We adjusted the results section on p.7, ll. 230-233 and the methods section on p.23, ll. 858-860:

A post-hoc repeated-measure ANOVA on alpha power changes (merged over Pz and Cz electrodes) with PP (high vs. low) and presentations (2 to 3) as within-subjects factors revealed a main effect of PP (F(1,32) = 5.42, p = 0.03, η2 = 0.15), and a significant interaction (F(1,32) = 7.38, p = 0.01, η2 = 0.19; Fig. 2e).

After confirming the existence of a significant cluster, we conducted an additional post-hoc repeated-measure ANOVA with averaged values of the identified time and frequency range of interest and merged over the Pz and Cz electrodes (see Fig. 2e).

Comment 46:(7) Figure 3: To better illustrate within- vs. between-subjects comparisons and promote transparency, please add individual points and lines between the within-subjects conditions.

According to this recommendation, we changed Figure 3 to add the individual data points by lines.

We modified Figure 3 on p.9, ll. 299-303:

Comment 47:(8) For the SW density time-bin analyses, please include statistics for all comparisons (i.e., through 0 s to 3 s) and say whether these were corrected for multiple comparisons.

According to this recommendation, we included now statistics for all comparisons.

We added table S6 table to the supplementary data on p.29, l.962:

Comment 48:(9) Consider reporting effect sizes.

We thank the reviewer for this recommendation and we added now effect sizes of significant results.

Comment 49:(10) For transparency and replicability, consider including a list of the four stimulus sets including their phoneme and biphone probabilities.

We included a list of the four stimulus sets with their phoneme and biphone probabilities

We added table S3 and table S4 to the supplementary data on pp. 26-27:

References

Asfestani MA, Brechtmann V, Santiago J, Peter A, Born J, Feld GB. 2020. Consolidation of Reward Memory during Sleep Does Not Require Dopaminergic Activation. J Cogn Neurosci 32:1688– 1703. doi:10.1162/JOCN_A_01585

Batterink LJ, Oudiette D, Reber PJ, Paller KA. 2014. Sleep facilitates learning a new linguistic rule.

Neuropsychologia 65:169–79. doi:10.1016/j.neuropsychologia.2014.10.024

Batterink LJ, Paller KA. 2017. Sleep-based memory processing facilitates grammatical generalization: Evidence from targeted memory reactivation. Brain Lang 167:83–93. doi:10.1016/J.BANDL.2015.09.003

Bohn OS, Best CT. 2012. Native-language phonetic and phonological influences on perception of American English approximants by Danish and German listeners. J Phon 40:109–128. doi:10.1016/J.WOCN.2011.08.002

Cairney SA, Guttesen A á. V, El Marj N, Staresina BP. 2018. Memory Consolidation Is Linked to Spindle-Mediated Information Processing during Sleep. Curr Biol 28:948-954.e4. doi:10.1016/j.cub.2018.01.087

Eberhard DM, Simons GF, Fennig CD. 2019. Ethnologue: Languages of the world . SIL International. Online version: http://www.ethnologue.com.

Fischer S, Born J. 2009. Anticipated reward enhances offline learning during sleep. J Exp Psychol Learn Mem Cogn 35:1586–1593. doi:10.1037/A0017256

Green DM, Swets JA. 1966. Signal detection theory and psychophysics., Signal detection theory and psychophysics. Oxford, England: John Wiley.

Griffiths B, Mazaheri A, Debener S, Hanslmayr S. 2016. Brain oscillations track the formation of episodic memories in the real world. Neuroimage 143:256–266. doi:10.1016/j.neuroimage.2016.09.021

Griffiths BJ, Martín-Buro MC, Staresina BP, Hanslmayr S, Staudigl T. 2021. Alpha/beta power decreases during episodic memory formation predict the magnitude of alpha/beta power decreases during subsequent retrieval. Neuropsychologia 153. doi:10.1016/j.neuropsychologia.2021.107755

Griffiths BJ, Mayhew SD, Mullinger KJ, Jorge J, Charest I, Wimber M, Hanslmayr S. 2019a. Alpha/beta power decreases track the fidelity of stimulus specific information. Elife 8. doi:10.7554/eLife.49562

Griffiths BJ, Parish G, Roux F, Michelmann S, van der Plas M, Kolibius LD, Chelvarajah R, Rollings DT, Sawlani V, Hamer H, Gollwitzer S, Kreiselmeyer G, Staresina B, Wimber M, Hanslmayr S. 2019b. Directional coupling of slow and fast hippocampal gamma with neocortical alpha/beta oscillations in human episodic memory. Proc Natl Acad Sci U S A 116:21834–21842. doi:10.1073/pnas.1914180116

Hanslmayr S, Spitzer B, Bäuml K-H. 2009. Brain oscillations dissociate between semantic and nonsemantic encoding of episodic memories. Cereb Cortex 19:1631–40. doi:10.1093/cercor/bhn197

Iber C, Ancoli‐Israel S, Chesson AL, Quan SF. 2007. The AASM Manual for the Scoring of Sleep and Associated Events: Rules, Terminology and Technical Specifications. Westchester, IL: American Academy of Sleep Medicine.

Klaassen AL, Heiniger A, Sánchez PV, Harvey MA, Rainer G. 2021. Ventral pallidum regulates the default mode network, controlling transitions between internally and externally guided behavior. Proc Natl Acad Sci U S A 118:1–10. doi:10.1073/pnas.2103642118

Lansink CS, Goltstein PM, Lankelma J V., McNaughton BL, Pennartz CMA. 2009. Hippocampus leads ventral striatum in replay of place-reward information. PLoS Biol 7. doi:10.1371/JOURNAL.PBIO.1000173

Luef EM, Resnik P. 2023. Phonotactic Probabilities and Sub-syllabic Segmentation in Language

Learning. Theory Pract Second Lang Acquis 9:1–31. doi:10.31261/TAPSLA.12468

Michelmann S, Bowman H, Hanslmayr S. 2016. The Temporal Signature of Memories: Identification of a General Mechanism for Dynamic Memory Replay in Humans. PLoS Biol 14:e1002528. doi:10.1371/journal.pbio.1002528

Proskovec AL, Heinrichs-Graham E, Wilson TW. 2019. Load Modulates the Alpha and Beta Oscillatory Dynamics Serving Verbal Working Memory. Neuroimage 184:256. doi:10.1016/J.NEUROIMAGE.2018.09.022

Reber AS. 1967. Implicit learning of artificial grammars. J Verbal Learning Verbal Behav 6:855–863.

doi:10.1016/S0022-5371(67)80149-X

Schreiner T, Rasch B. 2015. Boosting vocabulary learning by verbal cueing during sleep. Cereb Cortex 25:4169–4179. doi:10.1093/cercor/bhu139

Sterpenich V, van Schie MKM, Catsiyannis M, Ramyead A, Perrig S, Yang H-D, Van De Ville D, Schwartz S. 2021. Reward biases spontaneous neural reactivation during sleep. Nat Commun 2021 121 12:1–11. doi:10.1038/s41467-021-24357-5

Tamminen J, Lambon Ralph MA, Lewis PA. 2013. The role of sleep spindles and slow-wave activity in integrating new information in semantic memory. J Neurosci 33:15376–15381. doi:10.1523/JNEUROSCI.5093-12.2013

Tamminen J, Payne JD, Stickgold R, Wamsley EJ, Gaskell MG. 2010. Sleep spindle activity is associated with the integration of new memories and existing knowledge. J Neurosci 30:14356–60. doi:10.1523/JNEUROSCI.3028-10.2010

Zhu Y, Wang Q, Zhang L. 2021. Study of EEG characteristics while solving scientific problems with different mental effort. Sci Rep 11. doi:10.1038/S41598-021-03321-9